# RAIN-Merging: A Gradient-Free Method to Enhance Instruction Following in Large Reasoning Models with Preserved Thinking Format

**Zhehao Huang**[1] **Yuhang Liu**[1] **Baijiong Lin**[2] **Yixin Lou**[1] **Zhengbao He**[1] **Hanling Tian**[1]
**Tao Li**[1] **Xiaolin Huang**[1,3]

[1]Institute of Image Processing and Pattern Recognition, School of Automation and Intelligent Sensing, Shanghai Jiao Tong University

[2]The Hong Kong University of Science and Technology (Guangzhou)

[3]MoE Key Laboratory of System Control and Information Processing (Shanghai)

`{kinght_h,yuhangliu,loulou_0727,lstefanie,hanlingtian}@sjtu.edu.cn`
`{li.tao,xiaolinhuang}@sjtu.edu.cn`
`{blin241}@connect.hkust-gz.edu.cn`

## Abstract

Large reasoning models (LRMs) excel at a long chain of reasoning but often fail to faithfully follow instructions regarding output format, constraints, or specific requirements. We investigate whether this gap can be closed by integrating an instruction-tuned model (ITM) into an LRM. Analyzing their differences in parameter space, namely task vectors, we find that their principal subspaces are nearly orthogonal across key modules, suggesting a lightweight merging with minimal interference. However, we also demonstrate that naïve merges are fragile because they overlook the output format mismatch between LRMs (with explicit `thinking` and `response` segments) and ITMs (answers-only). We introduce **RAIN-Merging** (Reasoning-Aware Instruction-attention guided Null-space projection Merging), a gradient-free method that integrates instruction following while preserving thinking format and reasoning performance. First, with a small reasoning calibration set, we project the ITM task vector onto the null space of forward features at thinking special tokens, which preserves the LRM's structured reasoning mechanisms. Second, using a small instruction calibration set, we estimate instruction attention to derive module-specific scaling that amplifies instruction-relevant components and suppresses leakage. Across four instruction-following benchmarks and nine reasoning & general capability benchmarks, RAIN-Merging substantially improves instruction adherence while maintaining reasoning quality. The gains are consistent across model scales and architectures, translating to improved performance in agentic scenarios. Code is available at https://github.com/K1nght/RAIN-Merging.

## 1 Introduction

In the current boom of research, Large Reasoning Models (LRMs, like OpenAI-o1 (Jaech et al., 2024), DeepSeek-R1 (Guo et al., 2025)) have shown strong potential on tasks that require rigorous multi-step reasoning (Wei et al., 2022), such as mathematical derivation (Shao et al., 2024) and program synthesis (Guo et al., 2024). However, a discouraging paradox has emerged: although LRMs perform well in purely reasoning-oriented settings, they lag in instruction following (Fu et al., 2025a; Li et al., 2025a). They often generate lengthy logical derivations yet ignore user-specified formats, constraints, or specific operational requirements in the final response. This inconsistency undermines LRM practicality and reliability in real-world applications (Chkirbene et al., 2024), especially in agent (Qi et al., 2025) and professional tool deployments (Zhao et al., 2024).

A straightforward remedy is to continue training LRMs with supervised fine-tuning (SFT) to strengthen instruction following. However, building high-quality supervision datasets for tasks that require generating long chains of thought entails substantial annotation and computational re-

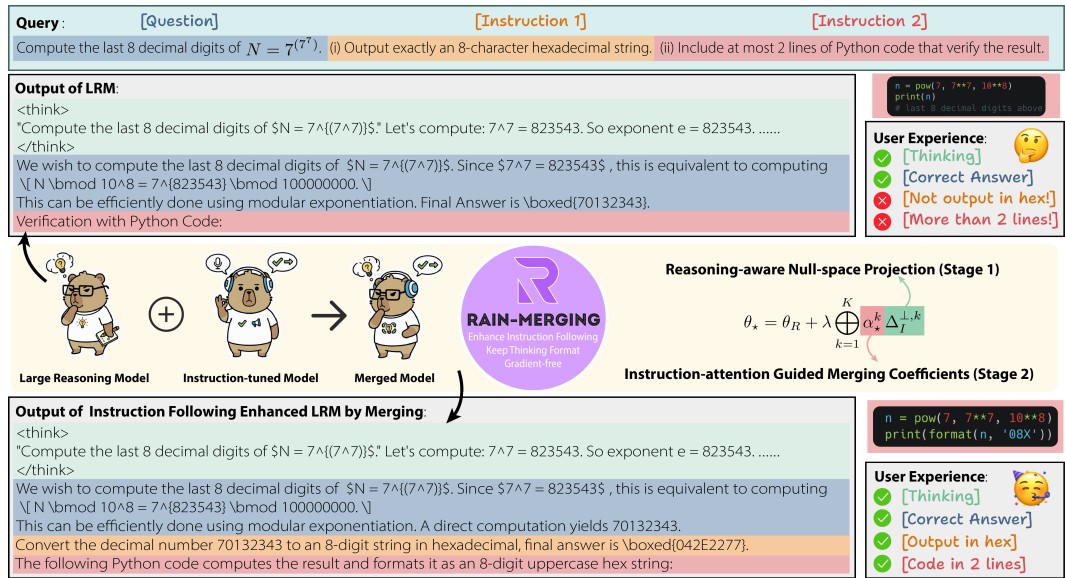

Figure 1: An overview of **RAIN-Merging**. In the case, the LRM arrives at the correct solution but ignores the required format and specific code. To preserve the reasoning structure, we perform training-free merging by combining a task vector projected onto the null space of the thinking format with instruction-attention guided coefficients. The merged model remains correct while satisfying the specified constraints. See **Sec.** 3 for details.

sources (Qin et al., 2025). Moreover, these post-training methods often induce capability regressions, with degradation in generality and in responses to unseen instructions (Shenfeld et al., 2025). In contrast, a training-free and compute-light alternative is model merging, which extracts parameter differences between fine-tuned and pre-trained models (namely the **task vector**), then combines these task vectors to create a unified model that preserves pre-trained knowledge while incorporating capabilities from multiple tasks (Ilharco et al., 2023). This motivates a central question: *whether we can merge the LRM and the Instruction-tuned Model (ITM) to enhance the instruction following while preserving its reasoning capability*.

We begin with a parameter-space analysis of the task vectors from the LRM and the Instruction-tuned Model (ITM) relative to their shared base. We find that their principal subspaces are nearly orthogonal across key modules, which indicates minimal interference between the two capabilities and suggests that merging is a promising lightweight way to enhance the LRM's instruction following (Ortiz-Jiménez et al., 2023). However, direct merging carries risks. LRMs and ITMs differ fundamentally in output structure: the former explicitly separates "thinking" and "response" with special markers (e.g., R1-style `<think>...</think>`), whereas the latter provides only a final answer. Traditional data-free merging (Ilharco et al., 2023; Goddard et al., 2024) prunes or scales the task vector purely from parameter-internal statistics to balance domain performance, thereby ignoring output-distribution mismatches and disrupting the LRM's structured reasoning. Recent work (Nobari et al., 2025; Yao et al., 2025; Chopra et al., 2025) has tried to guide merging with forward activations using small calibration sets. Although this introduces data-driven constraints, the lack of an explicit notion of the output mismatch between the two types still makes it difficult to achieve a stable and effective balance between preserving reasoning structure and improving instruction following.

To this end, we propose a two-stage merging strategy that enhances instruction-following capability without sacrificing the thinking format and reasoning performance of the LRM. First, leveraging task-vector orthogonality between the LRM and ITM, we preserve reasoning ability and enforce thinking-format invariance by projecting the ITM task vector into the null space derived from forward features at thinking tokens on a small reasoning-calibration set. This keeps the merged model's reasoning representations aligned with the original LRM and retains structured outputs. Second, while keeping these invariances fixed, we aim to enhance instruction-following performance as much as possible. We improve instruction adherence by estimating per-module importance based on attention outputs over instruction-related spans from a small set of instruction examples. Attention-guided coefficients are then assigned to strengthen instruction-relevant behaviors.We refer to the overall two-stage approach as Reasoning-Aware Instruction-guided Null-space projection Merging (**RAIN-Merging**) in **Fig.** 1, which effectively synergizes reasoning and instruction-following performance.

We conduct a systematic evaluation of our proposed method on four instruction-following benchmarks and on nine evaluation benchmarks that cover mathematics, code, STEM, and creative-writing capabilities. The results show that RAIN-Merging not only substantially improves the LRM's instruction-following ability but also maintains reasoning and general capability. Moreover, our method exhibits consistent stability across different model sizes and architectures, and demonstrates enhanced performance in agentic scenarios.

## 2 PRELIMINARY AND OBSERVATIONS

**Task Vector.** A task vector (Ilharco et al., 2023) characterizes the parameter delta from a base model to a task-specific one. A straightforward way to combine capabilities is **task arithmetic**, which linearly adds such deltas to a base model to obtain a multi-skilled model. This simple approach can work when tasks are compatible. However, for distinct abilities such as reasoning and instruction-following that impose different output structures (Yadav et al., 2023), naive linear addition may cause capability interference and disrupt the representations essential to each domain.

**Orthogonality between Reason & Instruction Task Vectors.** To examine whether capability interference arises when merging ITM $\theta_I$ into LRM $\theta_R$, we take the shared base model $\theta_B$ as reference and define the LRM task vector $\Delta_R = \theta_R - \theta_B$ and the ITM task vector $\Delta_I = \theta_I - \theta_B$. We perform singular value decomposition (SVD) within the main forward modules, e.g. attention and FFN, for these two task vectors and evaluate the principal subspace cosine similarity of their principal subspaces. As shown in **Fig. 2**, A1, A2, the two are nearly orthogonal since their similarities are all $< 0.1$. Prior studies (Ortiz-Jiménez et al., 2023) indicate that this phenomenon reflects a low degree of coupling between reasoning ability and instruction following in parameter space, which suggests that lightweight task-vector merging strategies can enhance instruction following while preserving the original reasoning performance. More details are in Appendix E.1.

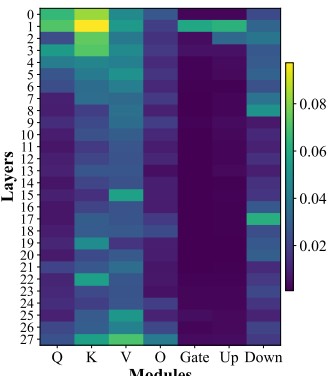

Figure 2: Principal subspace cosine similarity between LRM and ITM task vectors for each layer and submodule. The similarities are consistently low ($< 0.1$).

**Risks in Thinking Format During Merging.** However, orthogonality in parameter space is not sufficient to guarantee that the merged model will retain the LRM's structured output behavior, since this behavior is determined by downstream propagation and forward features (see Appendix E.1 for proof). In particular, the LRM relies on special tokens such as `<think>` and `</think>` to explicitly separate the reasoning segment from the answer segment, and these tokens are crucial in instruction-following tasks. For example, if the model fails to generate the terminator correctly after merging (as **Fig. 3**), it may conflate the reasoning content with the instruction-compliant response, which can violate constraints such as limits on output length. Therefore, although task-vector orthogonality suggests minimal capability interference, we still need to explicitly constrain the distributional shift of the output structure during merging to preserve the integrity of the reasoning process.

## 3 OUR RAIN-MERGING METHOD

**Notation.** For notational convenience in later derivations, we flatten model submodules by layer and head with index $k = 1, \ldots, K$ as $\theta = \bigoplus_{k=1}^{K} W^k := [\text{vec}(W^1)^\top, \ldots, \text{vec}(W^K)^\top]^\top$, where $\bigoplus$ denotes the block-wise concatenation that assembles disjoint parameter blocks into a single coordinate vector. More details of the forward mechanism in Transformer (Vaswani et al., 2017) are in Appendix G.1. Let $h_t^k$ denote the forward input vector at the $k$-th submodule and the $t$-th sampled token position. The corresponding linear map of this submodule admits the Kronecker-vectorization form (Koning et al., 1991) with Kronecker product $\otimes$, identity matrix $\text{diag}(1)$, and vectorization operator $\text{vec}(\cdot)$, as $W^k h_t^k = ((h_t^k)^\top \otimes \text{diag}(1)) \text{vec}(W^k)$. Stacking all sampled positions $t$ row-wise yields the forward feature operator $\Phi_{\{t\}}^k$ and outputs for the $k$-th submodule:

$$\Phi_{\{t\}}^k := \left[ (h_1^k)^\top \otimes \text{diag}(1), \ldots, (h_T^k)^\top \otimes \text{diag}(1) \right], \quad W^k h^k = \Phi_{\{t\}}^k \text{vec}(W^k). \quad (1)$$

**Optimization Objective.** To preserve the original reasoning performance of the LRM as much as possible, we take the reasoning model parameters $\theta_R$ as the anchor. We transform the ITM task vector $\Delta_I$ through a merging function $f$ to obtain $\Delta = f(\Delta_I)$, and form the merged model $\theta = \theta_R + \Delta$. Our goal is *to enhance instruction following without damaging the LRM's thinking format and reasoning performance*. We therefore formulate a constrained optimization problem: over the instruction data distribution $\mathcal{D}_I$, maximize the surrogate objective for instruction following, $\mathcal{J}_I(\theta) \triangleq \mathbb{E}_{x \sim \mathcal{D}_I} \mathbb{E}_{y \sim \pi_\theta(\cdot|x)} \big[ \mathrm{IF}(x, y) \big]$, while, over the reasoning data distribution $\mathcal{D}_R$, constraining the deviation between the model's output distribution within the segment of thinking special tokens $\Omega_{\text{think}}$ and the reference policy of the original reasoning model $\theta_R$. This constraint is quantified by aggregating the per-step KL divergence within the segment:

$$\mathcal{L}_{\text{think}}(\theta) \triangleq \mathbb{E}_{x \sim \mathcal{D}_R} \mathbb{E}_{y \sim \pi_{\theta_R}(\cdot|x)} \sum_{t \in \Omega_{\text{think}}(x)} \mathrm{KL}\big( \pi_\theta(\cdot \mid x, y_{<t}) \,\big\|\, \pi_{\theta_R}(\cdot \mid x, y_{<t}) \big). \tag{2}$$

The overall objective with tolerance $\delta$ is then:

$$\max_\Delta \ \mathcal{J}_I(\theta_R + \Delta) \quad \text{s.t.} \quad \mathcal{L}_{\text{think}}(\theta_R + \Delta) \leq \delta. \tag{3}$$

Noting that $\mathcal{J}_I$ is a surrogate objective for instruction following, referring to a class of functions IF that evaluate instruction alignment. In later we instantiate it with metrics based on instruction-attention alignment or leakage. In addition, motivated by the orthogonality between the LRM and ITM task vectors discussed earlier, we constrain only the conditional distribution in the segment of thinking special tokens and do not restrict the content generated in the other thinking or response segments, which preserves flexibility for improving instruction-following performance.

**Reasoning-aware Null-space Projection (Stage 1).** To satisfy the KL constraint on the segment of thinking special tokens, we try to seek a parameter subspace that preserves the thinking format. Intuitively, if we view the forward inputs at the thinking positions as a "measurement" of the reasoning style, then any parameter perturbation that is unresponsive under this measurement will not change the model's thinking pattern. This idea corresponds to projecting the perturbation onto the **null space** (Wang et al., 2021) of the forward feature operator $\Phi = \mathrm{blkdiag}\big(\Phi^1, \ldots, \Phi^K\big)$ (blkdiag denotes the block-diagonal matrix), namely $\mathcal{N}(\Phi) = \{v : \Phi v = 0\}$, as illustrated in **Fig. 3** (a). Such a null space projection keeps the token-level forward features at the thinking positions invariant. Formally, for each submodule $k$, we construct the least-squares orthogonal projector $P^\perp(\cdot)$ using the forward feature operator $\Phi^k_{\Omega_{\text{think}}}$ built from thinking special token indexs $\Omega_{\text{think}}$ to form the null space:

$$P^\perp\big(\Phi^k_{\Omega_{\text{think}}}\big) = \mathrm{diag}(1) - \Phi^k_{\Omega_{\text{think}}}{}^\top \Big( \Phi^k_{\Omega_{\text{think}}} \Phi^k_{\Omega_{\text{think}}}{}^\top \Big)^+ \Phi^k_{\Omega_{\text{think}}}, \tag{4}$$

where $(\cdot)^+$ denotes Moore-Penrose pseudoinverse. And then project the ITM submodule task vector $\Delta^k_I$ by submodule-wise and stack them to form the overall projected task vector to satisfy the null space constraint:

$$\mathrm{vec}\Big( \Delta^{\perp,k}_I \Big) = P^\perp(\Phi^k_{\Omega_{\text{think}}}) \, \mathrm{vec}\big(\Delta^k_I\big) \ \Rightarrow \ \Phi_{\Omega_{\text{think}}} \, \mathrm{vec}\big(\Delta^\perp_I\big) = 0, \ \text{where } \Delta^\perp_I = \bigoplus_{k=1}^K \Delta^{\perp,k}_I. \tag{5}$$

This projection keeps the merged model's intermediate representations and even the final logits at the thinking special tokens close to those of the anchor model. To verify its effectiveness in preserving the thinking format, we analyze a second-order expansion of the softmax KL divergence and show that the task vector after null-space projection satisfies the KL constraint on the special token output distribution in **Eq. (2)**. This yields the following **Prop. 1** (proof is in Appendix E.2):

**Proposition 1.** *Let the logits of sample $x$ at thinking special tokens $t \in \Omega_{think}(x)$ be $z_\theta(x, t)$, and let $\pi_\theta(\cdot \mid x, y_{<t}) = \mathrm{softmax}(z_\theta(x, t))$. By a second-order approximation of the softmax–KL divergence with a uniform upper bound, for any perturbation $u$,*

$$\mathrm{KL}\big( \mathrm{softmax}(z + u) \,\|\, \mathrm{softmax}(z) \big) \leq \tfrac{1}{8} \|u\|_2^2 + O\big(\|u\|_2^3\big). \tag{6}$$

*Assuming the model's intermediate representations are Lipschitz continuous and bounded, there exist constants $C_1, C_2 > 0$ such that for $u(x, t) = z_{\theta_R + \Delta}(x, t) - z_{\theta_R}(x, t)$, we have:*

$$\|u(x, t)\|_2 \leq C_1 \big\| \Phi \, \mathrm{vec}(\Delta) \big\|_2 + C_2 \|\Delta\|_2^2. \tag{7}$$

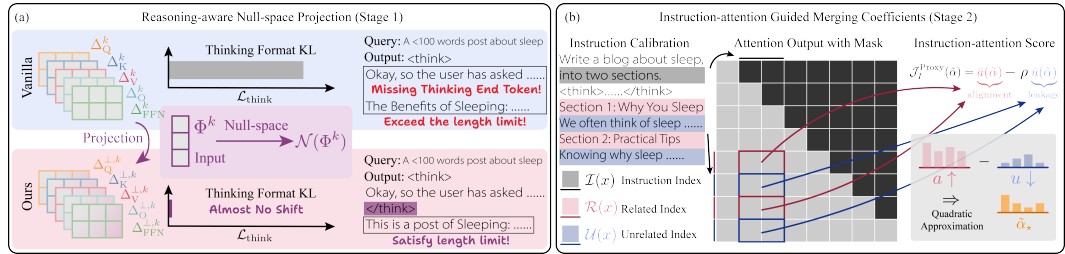

Figure 3: Two stages of our **RAIN-Merging** pipeline. (a) For each submodule, the ITM task vector is projected onto the null space preventing shifts in thinking format. (b) Given the instruction calibration set, we compute the instruction-attention score from attention outputs to obtain merging coefficients.

*Substituting the projected vector $\Delta_I^\perp = \bigoplus_{k=1}^K \Delta_I^{\perp,k}$ and the condition $\Phi \, \mathrm{vec}\big(\Delta_I^\perp\big) = 0$ yields:*

$$\mathcal{L}_{think}\big(\theta_R + \Delta_I^\perp\big) \le \tfrac{1}{8}\,\mathbb{E}_{x,t}\big[\|u(x,t)\|_2^2\big] + O\big(\mathbb{E}_{x,t}\|u(x,t)\|_2^3\big) = O\Big(\big\|\Delta_I^\perp\big\|_2^2\Big) \approx 0. \qquad (8)$$

Therefore, null-space projection in **Eq. (5)** approximately removes the thinking format constraint in the original objective and reduces the original optimization objective **Eq. (3)** to:

$$\max_{\Delta^\perp} \; \mathcal{J}_I\big(\theta_R + \Delta^\perp\big), \; \text{where } \Delta^\perp = f(\Delta_I^\perp). \qquad (9)$$

With the thinking-format constraint relaxed, we next focus on strengthening the task vector's effect on instruction following.

**Instruction-attention Guided Merging Coefficients (Stage 2).** To enhance the performance gain of the ITM task vector during merging, we seek a suitable gradient-free surrogate objective to instantiate $\mathcal{J}_I$. Prior studies (Guardieiro et al., 2025) suggest that failures in instruction following often stem from insufficient conditioning on the instruction span during decoding: *attention does not sufficiently focus on instruction-relevant tokens and instead leaks to unrelated regions*. A simple remedy is to amplify attention outputs on the instruction span at decoding time, which can remarkably improve instruction following. This approach, however, requires pre-identifying the instruction span, and excessive amplification may cause the model to ignore other necessary content. Motivated by this, we hypothesize that different layers and heads exhibit heterogeneous response behavior to instructions. Consequently, on the null-space–projected task vector $\Delta_I^{\perp,k}$, we introduce per-module scaling coefficients $\alpha = \{\alpha^k\} \in \mathbb{R}_+^K$ and reparameterize the merged model as $\theta(\alpha) = \theta_R + \bigoplus_{k=1}^K \alpha^k \Delta_I^{\perp,k}$ to instantiate merging function $f$. Given that attention outputs are directly coupled to the self-attention mechanism, we first focus on the merging coefficients of these submodules, as $\tilde{\alpha} = \{\alpha^{\tilde{k}}\} \in \mathbb{R}_+^{\tilde{K}}$, where $\tilde{k}$ denotes the self-attention submodule index. Our central intuition is that *an ideal merge should yield stronger attention responses on instruction-relevant spans (high alignment) while maintaining low attention activation on instruction-irrelevant content (low leakage)*. To translate this intuition into measurable quantities, we formalize the model's forward computation as follows and in **Fig. 3** (b). Let $\mathrm{Att}^{\tilde{k}}(x,\tilde{\alpha})[t,\tau]$ denote the attention output of the merged model with $\tilde{\alpha}$ at head $\tilde{k}$ from token position $t$ to $\tau$. For an instruction-following sample $x \sim \mathcal{D}_I$, we define the per-sample normalized alignment $a$ and leakage $u$ metrics for head $\tilde{k}$:

$$\underbrace{a^{\tilde{k}}(x,\tilde{\alpha})}_{\text{alignment}} := \sum_{t \in \mathcal{R}(x)} \sum_{\tau \in \mathcal{I}(x)} \frac{\mathrm{Att}^{\tilde{k}}(x,\tilde{\alpha})[t,\tau]}{|\mathcal{I}(x)|\,|\mathcal{R}(x)|}, \; \underbrace{u^{\tilde{k}}(x,\tilde{\alpha})}_{\text{leakage}} := \sum_{t \in \mathcal{U}(x)} \sum_{\tau \in \mathcal{I}(x)} \frac{\mathrm{Att}^{\tilde{k}}(x,\tilde{\alpha})[t,\tau]}{|\mathcal{I}(x)|\,|\mathcal{U}(x)|}. \qquad (10)$$

where $\mathcal{I}(x) \subset \{1, \dots, T\}$ represents the index set of instruction tokens that encodes the task description, formatting rules, constraints, and any examples in the query span. Likewise, $\mathcal{R}(x)$ denotes the set of output tokens whose content is directly constrained by the instruction in the response span, and $\mathcal{U}(x)$ the set of output tokens unrelated to the instruction. Taking expectations over instruction-following samples $\mathcal{D}_I$ and heads $\tilde{k}$ yields averaged alignment $\bar{a}(\tilde{\alpha}) = \sum_{\tilde{k}} \mathbb{E}_{x \sim \mathcal{D}_I}[a^{\tilde{k}}(x,\tilde{\alpha})]$ and averaged leakage $\bar{u}(\tilde{\alpha}) = \sum_{\tilde{k}} \mathbb{E}_{x \sim \mathcal{D}_I}[u^{\tilde{k}}(x,\tilde{\alpha})]$. We seek merging coefficients that achieve *high alignment* and *low leakage*. Accordingly, we combine the two metrics into a single **instruction-attention score** $\mathcal{J}_I^{\mathrm{Proxy}}$ with trade-off hyperparameter $\rho > 0$, instantiating the surrogate objective in

the reduced problem **Eq. (9)** then yields:

$$\max_{\tilde{\alpha}} \mathcal{J}_I^{\text{Proxy}}(\tilde{\alpha}) := \bar{a}(\tilde{\alpha}) \; - \; \rho \bar{u}(\tilde{\alpha}). \tag{11}$$

**Quadratic Approximation of Instruction-attention Score.** Although this objective is differentiable and could be optimized by gradient descent, we adopt a forward-pass approximation to reduce computation. Initialize at the directly merged point after projection, $\tilde{\alpha}_{(0)} \equiv \mathbf{1}$. Perform a second-order Taylor expansion of $\mathcal{J}_I^{\text{Proxy}}(\tilde{\alpha})$ around $\tilde{\alpha}_{(0)}$:

$$\mathcal{J}_I^{\text{Proxy}}(\tilde{\alpha}) \approx \mathcal{J}_I^{\text{Proxy}}(\tilde{\alpha}_{(0)}) + \nabla_{\tilde{\alpha}} \mathcal{J}_I^{\text{Proxy}}(\tilde{\alpha}_{(0)})^\top (\tilde{\alpha} - \tilde{\alpha}_{(0)}) + \tfrac{1}{2} (\tilde{\alpha} - \tilde{\alpha}_{(0)})^\top H (\tilde{\alpha} - \tilde{\alpha}_{(0)}), \tag{12}$$

where $H = \nabla_{\tilde{\alpha}}^2 \mathcal{J}_I^{\text{Proxy}}(\tilde{\alpha}_{(0)})$ is the Hessian. Writing $g = \nabla_{\tilde{\alpha}} \mathcal{J}_I^{\text{Proxy}}(\tilde{\alpha}_{(0)})$ and ignoring the constant term in **Eq. (11)**, we obtain the quadratic surrogate:

$$\mathcal{J}_I^{\text{quad}}(\tilde{\alpha}) = g^\top (\tilde{\alpha} - \tilde{\alpha}_{(0)}) + \tfrac{1}{2} (\tilde{\alpha} - \tilde{\alpha}_{(0)})^\top H (\tilde{\alpha} - \tilde{\alpha}_{(0)}). \tag{13}$$

① For first-order term $g$, if we restrict $\tilde{\alpha}$ to small deviations near $\tilde{\alpha}_{(0)}$ and adopt a linear approximation of alignment and leakage on merging coefficients, the per-head gradient can be estimated as:

$$g^{\tilde{k}} = \left. \frac{\partial \mathcal{J}_I^{\text{Proxy}}(\tilde{\alpha})}{\partial \tilde{\alpha}^{\tilde{k}}} \right|_{\tilde{\alpha}_{(0)}} \approx \frac{\partial \bar{a}(\tilde{\alpha})}{\partial \tilde{\alpha}^{\tilde{k}}} - \rho \frac{\partial \bar{u}(\tilde{\alpha})}{\partial \tilde{\alpha}^{\tilde{k}}} \approx \mathbb{E}_{x \sim \mathcal{D}_I} \left[ a^{\tilde{k}}(x, \tilde{\alpha}_{(0)}) - \rho \, u^{\tilde{k}}(x, \tilde{\alpha}_{(0)}) \right], \tag{14}$$

which replaces partial derivatives with the current metric values. In practice, this approximately scales the contribution of each head to instruction versus non-instruction attention mass, consistent with the intuition behind attention amplification.

② For second-order term $H$, to avoid the cost of computing the Hessian for large models, we adopt a diagonal approximation that limits the step size, $\tilde{H}^{\tilde{k}} = \text{diag}(1) + \mathbb{E}_{x \sim \mathcal{D}_I}[u^{\tilde{k}}(x, \tilde{\alpha}_{(0)})]$, where the second term imposes a stronger quadratic penalty on heads with higher leakage, thereby limiting their amplification. Substituting the approximations into the quadratic objective, dropping $\tilde{\alpha}_{(0)}$ for simplicity, and constraining $\tilde{\alpha} \in [\tilde{\alpha}_l, \tilde{\alpha}_u]^{\tilde{K}}$ to prevent scaling instability, we obtain a closed-form solution to the convex quadratic program:

$$\max_{\tilde{\alpha} \in [\tilde{\alpha}_l, \tilde{\alpha}_u]^{\tilde{K}}} \left( g^\top \tilde{\alpha} - \tfrac{1}{2} \tilde{\alpha}^\top \tilde{H} \tilde{\alpha} \right) \quad \Rightarrow \quad \tilde{\alpha}_\star^{\tilde{k}} = \text{clip}_{[\tilde{\alpha}_l, \tilde{\alpha}_u]} \left( \frac{g^{\tilde{k}}}{\tilde{H}^{\tilde{k}}} \right), \tag{15}$$

where $\tilde{H} = \text{diag}(\tilde{H}^{\tilde{k}})$ and $\text{clip}_{[a,b]}(\cdot)$ clips to the interval $[a, b]$. Thus, by a second-order expansion with engineering approximations and using only forward attention statistics in a gradient-free manner, we approximate the optimal merging coefficients $\tilde{\alpha}_\star$ of self-attention submodules that increase instruction alignment while controlling attention leakage to non-instruction content. For modules shared across attention heads, such as the feed-forward network (FFN), we set the layer-wise coefficient to the average over heads. Aggregating the coefficients for all submodules yields the complete instruction attention guided merging coefficients $\alpha_\star = \{\alpha_\star^k\}$.

**Combined to Our Two-stage Merging Method.** We chain "Reasoning-aware Null-space Projection (Stage 1)" with "Instruction-attention Guided Merging Coefficients (Stage 2)" to propose a fully gradient-free merging pipeline, termed *Reasoning-Aware Instruction-attention guided Null-space projection Merging* (**RAIN-Merging**) as **Fig. 3**. Our method addresses the challenge in the original optimization problem of **Eq. (3)**, improving instruction following while preserving the reasoning structure after merging. The final merged model is:

$$\theta_\star = \theta_R + \lambda \bigoplus_{k=1}^K \alpha_\star^k \Delta_I^{\perp,k}, \tag{16}$$

where $\lambda$ is a global scaling coefficient that controls the merging strength. The entire procedure only relies on forward-feature extraction and attention statistics, and does not require gradient-based updates. RAIN-Merging offers a low-cost, interpretable path to strengthen instruction following in LRMs, filling the gap left by costly SFT.

Table 1: Comprehensive comparison of instruction following and reasoning & general capabilities. We merge Qwen2.5-7B-Instruct (ITM) into DeepSeek-R1-Distill-Qwen-7B (LRM) and compare our RAIN-Merging against multiple merging methods as well as SFT trained on the same calibration data. "Avg." denotes the average over all subsets. "RT" reports the run-time for merging or training in minutes. The best and second-best results are highlighted in **bold** and underlined, respectively.

| Method | Instruction Following | | | | | Reasoning & General | | | | | RT |
|---|---|---|---|---|---|---|---|---|---|---|---|
| | IFEval | CELLO | Info Bench | Complex Bench | Avg. | Math | GPQA | Aider | Arena-Hard-v2 | Avg. | |
| ITM | **70.43** | **19.15** | **78.49** | **43.63** | **52.92** | 47.27 | 29.80 | **33.33** | 62.86 | 43.32 | – |
| LRM | 55.45 | 16.59 | 71.73 | 32.72 | 44.12 | 64.75 | 44.44 | 29.63 | 65.29 | 51.03 | – |
| SFT | 62.48 | 17.11 | 68.58 | 32.15 | 45.08 | 62.57 | 41.92 | 28.89 | 64.67 | 49.51 | 120.32 |
| *Data-free Merging* | | | | | | | | | | | |
| Task Arithmetic | 60.44 | 16.97 | 73.07 | 33.34 | 45.96 | 64.22 | 42.93 | 26.67 | 64.53 | 49.59 | 0.93 |
| SLERP | 58.96 | 17.56 | 72.18 | 34.93 | 45.95 | 63.82 | 42.93 | 31.85 | 65.29 | 50.97 | 1.12 |
| Karcher | 62.11 | 17.99 | 73.16 | 34.06 | 46.83 | 64.85 | 48.99 | 30.37 | **66.13** | 52.58 | 1.20 |
| TIES | 58.60 | 18.48 | 73.91 | 34.40 | 46.35 | 65.44 | 46.46 | 32.59 | 63.47 | 51.99 | 1.18 |
| DARE-TIES | 60.81 | 17.88 | 73.33 | 33.49 | 46.38 | 64.26 | 47.98 | 29.63 | 64.17 | 51.51 | 2.21 |
| *Data-dependent Merging* | | | | | | | | | | | |
| ACM-TIES | 59.33 | 16.45 | 72.44 | 33.75 | 45.50 | 64.57 | 45.96 | 32.59 | 62.00 | 51.28 | 12.45 |
| LEWIS-TIES | 60.44 | 17.41 | 72.67 | 34.40 | 46.23 | 62.07 | 48.99 | 31.11 | 64.80 | 51.74 | 16.60 |
| AIM-TIES | 62.78 | 17.93 | 73.11 | 34.28 | 47.02 | 65.92 | 49.49 | **33.33** | 63.64 | 53.10 | 18.51 |
| RAIN-Merging | 63.22 | 19.03 | 74.53 | 35.66 | 48.11 | **68.75** | **54.55** | **33.33** | 65.73 | **55.59** | 20.96 |

**Implementation details.** To balance compute and storage efficiency, we merge only the core modules that are most sensitive to attention outputs, namely the Q, K, V, O, and FFN parameters. In Stage 1, we sample 150 examples from the Mixture-of-Thoughts (Face, 2025) dataset distilled from DeepSeek-R1 (Guo et al., 2025) from to form the reasoning calibration set. In Stage 2, we an instruction calibration set obtained by distilling DeepSeek-R1 on IFEval (Zhou et al., 2023b), followed by LLM-as-Judge filtering and manual screening, resulting a total of 365 samples. More details of implementation, complete algorithm pseudocode, calibration set construction, and ablation studies are provided in Appendix G, F, H, and J.3.

## 4 EXPERIMENTS

In this section, we empirically investigate three research questions:

- **RQ1 (Effectiveness and Efficiency).** Compared with baseline methods, can RAIN-Merging improve instruction-following while maintaining reasoning capabilities, maintaining the computational and memory efficiency characteristic of gradient-free approaches? (**Tab.** 1 and **Fig.** 4)
- **RQ2 (Scalability).** How well does RAIN-Merging scale across models of varying sizes and architectures, and does it perform effectively in interactive agentic scenarios? (**Tab.** 2, 3)
- **RQ3 (Ablation).** What roles do the two stages of RAIN-Merging play? Specifically, does Stage 1 preserve the format of thinking segments and the output distribution, and does Stage 2 enhance instruction-attention scores? (**Tab.** 4 and **Fig.** 5, **Fig.** 6)
- **RQ4 (Investigation).** How does augmenting the instruction following capabilities of LRMs influence their inherent reasoning efficacy? Specifically, how does this evolution in reasoning ability manifest in specific cases? (**Tab.** 5, 6, **Sec.** J.9, J.10)

### 4.1 EXPERIMENTAL SETUP

We begin with a brief overview of the benchmarks, models, and baselines used in our experiments. Additional details on experimental settings, benchmarks and evaluation metrics, and hyperparameters are provided in Appendix I.

**Benchmarks.** To comprehensively assess instruction following, we use four mainstream benchmarks: **IFEval** (Zhou et al., 2023b), **CELLO** (He et al., 2024), **InfoBench** (Qin et al., 2024), and **ComplexBench** (Wen et al., 2024). To comprehensively evaluate reasoning and general capabilities, we use nine benchmarks: Mathematical reasoning is evaluated by aggregating results from six benchmarks, as **Math**. We also measure performance on code editing (**Aider** (Aider, 2024)), STEM (**GPQA** (Rein et al., 2024)), and creative writing (**Arena-Hard-v2** (Li et al., 2024)) to reflect general

Table 2: Merging performance and relative gains of RAIN-Merging across model three scales and two architectures. We merge the corresponding ITM into the LRM with base models: Qwen2.5-1.5B/14B/32B, and Llama-3.1-8B. "Avg." denotes the average over all subsets. For each scale, the subsequent "*(relative gain)*" row reports the relative improvement of our method over the LRM, highlighted in green.

| Model | Instruction Following | | | | | Reasoning & General | | | | |
|---|---|---|---|---|---|---|---|---|---|---|
| | IFEval | CELLO | Info Bench | Complex Bench | Avg. | Math | GPQA | Aider | Arena-Hard-v2 | Avg. |
| Qwen2.5-1.5B-Instruct | 36.78 | 19.04 | 64.76 | 27.83 | 37.10 | 31.77 | 25.76 | 16.30 | 38.45 | 28.07 |
| DeepSeek-R1-Distill-Qwen-1.5B | 39.00 | 16.03 | 55.29 | 21.54 | 32.97 | 41.62 | 29.29 | 14.07 | 39.73 | 31.18 |
| Qwen2.5-1.5B-RAIN-Merging | 41.59 | 16.51 | 58.18 | 23.62 | 34.97 | 45.87 | 33.33 | 14.81 | 40.93 | 33.74 |
| *(relative gain)* | +6.64% | +2.98% | +5.23% | +9.63% | +6.09% | +10.21% | +13.79% | +5.26% | +3.02% | +8.20% |
| Llama-3.1-8B-Instruct | 68.58 | 27.21 | 78.67 | 38.47 | 53.23 | 35.59 | 25.25 | 34.07 | 72.23 | 41.79 |
| DeepSeek-R1-Distill-Llama-8B | 58.41 | 17.78 | 73.33 | 38.38 | 46.97 | 60.21 | 38.38 | 27.41 | 71.93 | 49.48 |
| Llama-3.1-8B-RAIN-Merging | 63.77 | 18.84 | 77.38 | 38.93 | 49.73 | 61.95 | 43.94 | 30.37 | 77.07 | 53.33 |
| *(relative gain)* | +9.18% | +5.99% | +5.52% | +1.42% | +5.86% | +2.89% | +14.47% | +10.81% | +7.15% | +7.78% |
| Qwen2.5-14B-Instruct | 79.85 | 20.13 | 83.38 | 44.19 | 56.89 | 52.73 | 36.87 | 37.04 | 74.40 | 50.29 |
| DeepSeek-R1-Distill-Qwen-14B | 71.35 | 18.71 | 81.33 | 40.68 | 53.02 | 72.31 | 57.07 | 33.33 | 80.67 | 60.85 |
| Qwen2.5-14B-RAIN-Merging | 76.71 | 19.57 | 84.13 | 44.63 | 56.26 | 74.58 | 57.58 | 40.00 | 86.25 | 64.60 |
| *(relative gain)* | +7.51% | +4.58% | +3.44% | +9.69% | +6.11% | +3.13% | +0.88% | +20.00% | +6.92% | +6.17% |
| Qwen2.5-32B-Instruct | 78.56 | 18.59 | 84.40 | 46.91 | 57.11 | 52.35 | 36.87 | 57.78 | 81.90 | 57.22 |
| DeepSeek-R1-Distill-Qwen-32B | 76.52 | 19.69 | 83.56 | 44.44 | 56.05 | 68.00 | 60.10 | 54.81 | 82.00 | 66.23 |
| Qwen2.5-32B-RAIN-Merging | 77.26 | 19.96 | 84.76 | 45.74 | 56.93 | 75.67 | 61.62 | 54.07 | 83.70 | 68.77 |
| *(relative gain)* | +0.97% | +1.39% | +1.44% | +2.93% | +1.57% | +11.28% | +2.52% | -1.35% | +2.07% | +3.83% |

and reasoning capabilities. For agentic scenarios, we use **ALFWorld** (Shridhar et al., 2021) and **WebShop** (Yao et al., 2022), two realistic multi-turn interactive tasks, to evaluate how well the model integrates reasoning and instruction following to solve complex problems.

**Models.** We evaluate RAIN-Merging on models of different sizes and architectures: DeepSeek-R1-Distill-Qwen-1.5B/7B/14B/32B (Guo et al., 2025) (LRM) and Qwen2.5-1.5B/7B[1]/14B/32B-Instruct (Yang et al., 2025) (ITM), as well as the Llama family (Dubey et al., 2024) using DeepSeek-R1-Distill-Llama-8B (LRM), its instruction-tuned counterpart Llama-3.1-8B-Instruct (ITM).

**Baselines.** We include several data-free, task-vector based merging methods: **Task Arithmetic** (Ilharco et al., 2023), **SLERP** (Biship, 2007; Goddard et al., 2024), **Karcher** (Nielsen & Bhatia, 2013; Goddard et al., 2024), **TIES** (Yadav et al., 2023), and **DARE** (Yu et al., 2024). We also compare with data-dependent, activation-based merging approaches that leverage small calibration sets, including **ACM** (Yao et al., 2025), **LEWIS** (Chopra et al., 2025), and **AIM** (Nobari et al., 2025). To strengthen baseline performance, we apply TIES on top of other merging baselines as in previous work (Wu et al., 2025). In addition, we report a training baseline using **SFT** on the same calibration data.

## 4.2 RESULTS

**Performance Comparison with Baseline Methods. (RQ1)** As shown in **Tab. 1**, RAIN-Merging achieves overall gains across both instruction-following and reasoning & general capability evaluations, outperforming all merging baselines. While Task Arithmetic and SFT can improve instruction following to some extent, they typically do so at the cost of reasoning and general capabilities. In contrast, our method consistently surpasses all baselines on instruction-following, mathematical reasoning, and general-capability benchmarks. Our merged LRM trails the ITM slightly on instruction following, indicating room for further improvement. Interstingly, the merged model exhibits stable gains in reasoning and general ability. We hypothesize that stronger instruction adherence improves the quality of the model's internal chain of thought, which yields better reasoning performance. Overall, RAIN-Merging substantially enhances instruction following without sacrificing the LRM's reasoning and general capabilities.

**Run-time and Memory Analysis. (RQ1)** Our method achieves a favorable efficiency trade-off. Its runtime, though slightly above activation-based merging baselines due to null-space computation, is far below SFT (RT in **Tab. 1**). Similarly, while storing hidden features increases memory use compared to other merging methods, its footprint remains much smaller than SFT's (**Fig. 4**). This demonstrates our approach as a highly practical, training-free alternative for enhancing LRMs.

---

[1]Although DeepSeek-R1-Distill-Qwen-1.5B/7B are trained from Qwen2.5-Math-1.5B/7B (Yang et al., 2024a), we find that Qwen2.5-Math-1.5B/7B-Instruct do not outperform the distilled LRMs in instruction following. We therefore select the stronger instruction followers, Qwen2.5-1.5B/7B-Instruct, as ITMs.

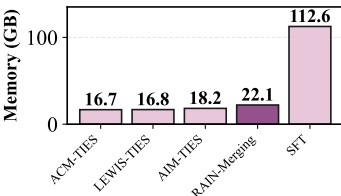

Figure 4: GPU memory usage comparison between different methods under the same configuration as **Tab. 1**.

Table 3: Performance of RAIN-Merging in agent settings. We merge Qwen2.5-7B-Instruct (ITM) into DeepSeek-R1-Distill-Qwen-7B (LRM).

| Model | ALFWorld | WebShop |
|---|---|---|
| ITM | 17.50 | 10.45 |
| LRM | 22.00 | 26.63 |
| RAIN-Merging | 25.00 | 29.42 |

Table 4: Performance of ablation on Stage 1 and Stage 2, under the same setup as **Tab. 1**. "I Avg." and "R Avg." denote the average performance on instruction-following and reasoning & general benchmarks.

| Method | I Avg. | R Avg. |
|---|---|---|
| RAIN-Merging w/o Stage 2 | 46.58 | 54.92 |
| RAIN-Merging w/o Stage 1 | 47.62 | 52.44 |
| RAIN-Merging | 48.11 | 55.59 |

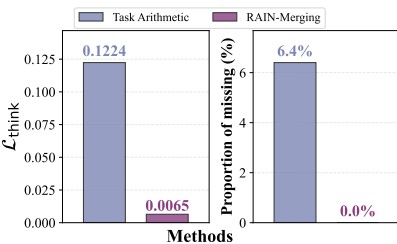

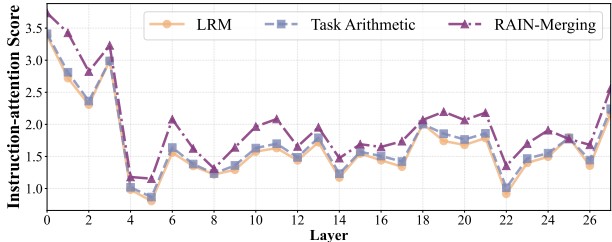

Figure 5: $\mathcal{L}_{\text{think}}$ in **Eq. (2)** (left) on the reasoning calibration validation set, and the proportion of generations missing the closing `</think>` token (right) on IFEval under the same configuration as **Tab. 1**.

Figure 6: Layer-wise instruction attention score (alignment − leakage). Per-layer scores on IFEval instruction examples; higher is better. We compare the unmerged LRM, Task Arithmetic, and RAIN-Merging when merging Qwen2.5-7B-Instruct (ITM) into DeepSeek-R1-Distill-Qwen-7B (LRM).

**Performance on Models of Different Sizes and Architectures. (RQ2)** To evaluate the scalability of our method across model sizes and architectures, we conduct experiments on several configurations, including the Qwen2.5 family distilled from DeepSeek-R1 at 1.5B, 14B, and 32B parameters, and the 8B model built on the Llama 3.1 architecture. As reported in **Tab. 2**, our method consistently enhances instruction-following and reasoning performance, achieving average improvements from 1.57% to 8.20% on LRMs. These results confirm that RAIN-Merging robustly strengthens instruction adherence and preserves complex reasoning across diverse model sizes and architectures.

**Performance in Agentic Scenarios. (RQ2)** To further assess the practical benefits of improved instruction following, we evaluate the merged model on two representative agentic scenarios, Web-Shop and AlfWorld. As shown in **Tab. 3**, the merged model achieves better performance than the original LRM and ITM in these scenarios, indicating that enhanced instruction understanding and reasoning effectively support multi-turn interaction and complex decision making. These results also demonstrate that our gradient-free approach is effective for increasing the real-world utility of LRMs.

**Ablation Study of Stage 1 and Stage 2. (RQ3)** We investigate the contribution of the two components in RAIN-Merging, shown in **Tab. 4**. When without Stage 2, the merged model retains reasoning and general capabilities while achieving competitive instruction-following performance. Conversely, when without Stage 1, instruction-following performance improves further but at a noticeable cost to reasoning and general ability, as it lacks protection of the thinking format. Incorporating both stages yields the best trade-off: Stage 1 ensures reasoning performance is maintained while Stage 2 boosts instruction-following performance. Thereby, both stages play critical and complementary roles.

**Effectiveness of Null-space Projection. (RQ3)** To assess how our null-space projection in Stage 1 preserves thinking formats, we evaluate its impact on thinking special token distributions and resulting generation outputs. We measure the KL divergence near thinking tokens as in **Eq. (2)** and the rate of missing </think> tokens. Results **Fig. 5** show that Task Arithmetic substantially alters the distribution ($\mathcal{L}_{\text{think}} = 0.1224$) and results in 6.4% missing </think> tokens, violating the output format. Our approach, in contrast, only induces minimal change ($\mathcal{L}_{\text{think}} = 0.0065$) and ensures no missing tokens (0.0%). These findings indicate that null-space projection successfully protects thinking formats.

**Effectiveness of Merging Coefficients. (RQ3)** To validate the merging coefficients, we compare the Instruction-Attention Score in **Eq. (11)** across layers before and after merging under different methods. As shown in **Fig. 6**, instruction-attention guided coefficients in Stage 2 enable **RAIN-Merging** to consistently outperform both the LRM and Task Arithmetic, exhibiting a higher alignment and lower leakage. This finding suggests that our weighted reparameterization of merging submodules enhances

Table 5: Merging performance of RAIN-Merging on MathIF under the same configuration as **Tab.** 1. **IF Acc.** and **Math Acc.** are the accuracy of instruction constraints and math answers, respectively. **Both Acc.** represents both constraints and math answers are correct.

Table 6: Evaluation of reasoning and answer traces under the same configuration as **Tab.** 1. We report Reasoning Internal Coherence (**RIC**) and Reasoning-Answer Alignment (**RAA**) on IFEval, AIME25, and GPQA (0-5 scale). The subsequent "*(relative gain)*" row reports the relative improvement of our method over the LRM, highlighted in green.

| Model | IF Acc. | Math Acc. | Both Acc. |
|---|---|---|---|
| ITM | 48.81 | 40.95 | 19.76 |
| LRM | 25.86 | 53.81 | 12.62 |
| RAIN-Merging | 35.10 | 54.76 | 20.48 |
| *(relative gain)* | +35.73% | +1.77% | +62.26% |

| Model | IFEval | | AIME25 | | GPQA | | Avg. |
|---|---|---|---|---|---|---|---|
| | RIC | RAA | RIC | RAA | RIC | RAA | |
| LRM | 4.58 | 4.41 | 4.50 | 3.60 | 3.53 | 3.76 | 4.06 |
| RAIN-Merging | 4.61 | 4.51 | 4.50 | 4.10 | 3.56 | 4.26 | 4.26 |
| *(relative gain)* | +0.77% | +2.26% | 0.00% | +13.89% | +0.86% | +13.31% | +4.78% |

activation along instruction-aware pathways while slightly suppressing leakage, which improves instruction following without altering the original reasoning pattern.

**Joint Evaluation of Reasoning and Instruction-Following. (RQ4)** To more directly assess whether RAIN-Merging can jointly maintain strong reasoning and strict instruction-following within a *single* task, we further evaluate on *MathIF* (Fu et al., 2025b). MathIF augments math problems with verifiable constraints and reports both constraint satisfaction and math correctness, as well as a joint metric requiring both. As shown in **Tab.** 5, RAIN-Merging markedly improves MathIF instruction hard accuracy over the LRM while leaving math correctness essentially unchanged. Crucially, on *Both Acc*, the merged model rises from $12.62\%$ to $20.48\%$ ($+62.26\%$ relative), outperforming both the LRM and the ITM, demonstrating a substantial gain on the core goal of being *both correct and follow* within a single benchmark.

**Reasoning and Answer Traces Evaluation. (RQ4)** To complement our reasoning and answer quality evaluations, we adopt the framework of Wang et al. (2025) and perform a process-level analysis of reasoning traces and answers. We measure two chain-of-thought level metrics by LLM-as-a-judge: ① *Reasoning Internal Coherence (RIC)* assesses how logically consistent and self-contained the reasoning trace is. ② *Reasoning–Answer Alignment (RAA)* measures how well the reasoning trace semantically supports the final answer. For each sample we provide the question, the ground-truth answer, the full reasoning trace, and the model's final answer response. The judge then assigns 0-5 scores for RIC and RAA. **Tab.** 6 shows that RAIN-Merging yields slight improvements in both RIC and RAA on IFEval. For the reasoning-focused benchmarks AIME25 and GPQA, it maintains RIC scores comparable to the LRM, while achieving a marked increase in RAA, notably by over 13% in both cases. In other words, the internal coherence of the reasoning is preserved, with the connection between the reasoning and the final decision becoming noticeably tighter.

**Case Study on Instruction Following and Reasoning. (RQ4)** To provide a granular understanding of how RAIN-Merging enhances model performance, we conducted a series of case studies focusing on instruction following and complex reasoning in Appendix J.9 and J.10. In instruction-following scenarios, RAIN-Merging effectively rectifies common failure modes of baseline LRMs, such as the violation of explicit formatting rules. On reasoning tasks, the merged model demonstrates superior internal consistency. It not only corrects localized numerical slips through more deliberate and step-by-step computations but also successfully bridges the "knowing-doing gap" (Schmied et al., 2025) where the baseline LRM's final selection contradicts its own correct internal derivation. These qualitative observations suggest that RAIN-Merging achieves more than just preserving reasoning chains, it actively aligns the model's decision-making process with its logical reasoning.

## 5 CONCLUSION

We propose RAIN-Merging, a gradient-free method to enhance instruction following in LRMs while preserving their structured reasoning outputs. By projecting the instruction task vector onto the null space of the thinking format and scaling it by instruction-attention guided coefficients, RAIN-Merging achieves a balance between instruction following and reasoning structure preservation. The method is evaluated on instruction-following, reasoning & general capability, agentic benchmarks, showing that RAIN-Merging not only substantially improves the LRM's instruction-following ability but also brings gains in reasoning and general capability across several settings.

ACKNOWLEDGMENTS

The authors would like to thank the anonymous reviewers for their insightful comments.

The author would also like to express his gratitude to Yucheng Bai for his assistance in identifying and correcting typographical errors within the formulas.

The resaerch leading to these results has received funding from the AI for Science Program, Shanghai Municipal Commission of Economy and Informatization (No. 2025-GZL-RGZN-BTBX-02026) and the National Natural Science Foundation of China (No. 62376155).

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

APPENDIX

## A  ETHICS STATEMENT

This research adheres to the licenses and applicable laws governing upstream open-source models and datasets. RAIN-Merging is developed using publicly available weights and data that permit derivation and redistribution.

**Safety.** Model merging can introduce "capability or safety drift," such as new biases, jailbreak risks, or shifts in hallucination patterns while improving instruction following. The merged model may produce inaccurate, biased, or inappropriate content. It must not be used directly in high-risk decision-making contexts such as medicine, law, or finance. Any production deployment should include human oversight, operation logging, rate limiting, and compliance review procedures.

**Dataset use.** We rely only on data authorized for academic reproducibility. During data cleaning, we make every effort to remove personally identifiable information and sensitive content. We also disclose potential dataset biases, coverage gaps, and risks of benchmark contamination.

**Societal impact.** We caution that generative models may exacerbate information asymmetries, reinforce stereotypes, or be applied to produce misleading content. We firmly oppose misuse and will work with the community to address any identified negative impacts.

## B  REPRODUCIBILITY STATEMENT

To ensure the reproducibility of our results, we provide the following resources and documentation: all algorithm implementations and experiment scripts will be released anonymously with the **supplementary materials**, accompanied by documentation of key functions and the project directory structure. The calibration datasets used in our experiments will be made available alongside the appendix. Public links are included for all open-source models and datasets used in this work.

## C  LLM USAGE STATEMENT

We used large language models (LLMs) in the following stages and disclose their roles as follows:

**Writing Stage.** LLMs (both closed- and open-source) were used only for copyediting and grammar checking, including terminology normalization, syntactic polishing, and formatting. They were not used to generate claims, collect evidence, or construct results.

**Benchmark Evaluation.** When a benchmark's original paper or community practice requires a closed-source LLM (for example, as a judge or as a baseline), we strictly follow the prescribed protocol and disclose the exact model versions.

**Calibration Set Construction.** We adopt an LLM-as-Judge procedure for automated filtering and scoring of candidate samples as an initial pass (producing only scores or labels; generated text is not used as a training target). A human second-pass review follows to ensure data quality and compliance. All third-party data and models are used within their licenses, with source links and permission details provided.

## D  RELATED WORK

**Large Reasoning Model.** Early studies show that prompting models to explicitly produce intermediate steps during reasoning can substantially improve complex reasoning performance, as in Chain-of-Thought (Wei et al., 2022) and Tree-of-Thought (Yao et al., 2023a). Building on this insight, a new generation of LRMs has shifted toward training paradigms that directly incentivize reasoning with reinforcement learning; for example, OpenAI's o1 series and DeepSeek-R1 report marked advances on tasks in mathematics and code that require extended reasoning (Jaech et al., 2024; Guo et al., 2025). These models typically generate structured "thought processes" or "thinking formats," yet in real applications they often exhibit tension with strict instruction following. Beyond explicit intermediate reasoning such as CoT and ToT, subsequent work further improves reasoning

quality and stability: Self-Consistency samples multiple solution paths and uses majority voting to increase reliability; Least-to-Most decomposes complex problems into subgoals ordered from easy to hard; Program-of-Thoughts and PAL externalize the reasoning into executable programs, decoupling computation from reasoning and substantially reducing arithmetic and procedural errors (Wang et al., 2023; Zhou et al., 2023a; Chen et al., 2022; Gao et al., 2023). In the "reasoning plus acting" paradigm, ReAct interleaves thought traces with tool interactions to mitigate hallucinations, while Reflexion employs language-based self-reflection and memory to iteratively refine policies over multi-turn interactions (Yao et al., 2023b; Shinn et al., 2023). In parallel, LRMs are trained with process-level feedback and reinforcement learning to directly encourage thinking before answering: the o1 system emphasizes large-scale RL and thinking-first training and alignment strategies, and DeepSeek-R1 shows that under weak or no supervision, pure RL (e.g., GRPO) can induce longer and more stable chains of thought (Jaech et al., 2024; Guo et al., 2025). Moreover, process supervision and process reward models (PRMs) provide finer-grained step-level feedback that, compared with outcome supervision of final answers, better cultivates verifiable reasoning chains and test-time expansion (Lightman et al., 2024). RAIN-Merging is complementary to this trajectory: instead of retraining the LRM, we preserve the thinking format at merge time and use structured coefficients to selectively enhance instruction responses, thereby striking a balance between fidelity to the reasoning structure and improved instruction following.

**Instruction Following.** In the alignment paradigm, InstructGPT systematically improved the stability of "following user intent" through reinforcement learning from human feedback (RLHF), and showed that small instruction-tuned models can achieve strong human preference scores, establishing a foundation for subsequent research on instruction following (Ouyang et al., 2022). For objective evaluation, IFEval targets programmatically verifiable rules, for example, length limits, keywords, formatting, to reduce subjective scoring noise and facilitate reproducibility and fair comparison (Zhou et al., 2023b). CELLO abstracts multi-dimensional attributes from real-world complex instructions, such as multi-step dependencies, format or quantity constraints, and semantic consistency, to characterize where LLMs struggle with complex instruction understanding (He et al., 2024). InfoBench proposes a decomposed metric that breaks a complex instruction into checkable sub-requirements, enabling finer-grained measurement of compliance and error sources (Qin et al., 2024). ComplexBench emphasizes the compositional challenge of multiple simultaneous constraints, systematically testing robustness and trade-offs when many constraints co-occur (Wen et al., 2024). Building on these mainstream benchmarks, this work introduces an instruction-attention–oriented merging strategy: during merging, we quantitatively constrain and amplify the model's responsiveness to instruction-relevant spans while maintaining the stability of its long-chain reasoning format, thereby balancing compliance and an interpretable process.

**Model Merging.** Parameter-space merging offers a training-free or low-data path for integrating capabilities. Model Soup averages weights from multiple fine-tuned checkpoints to improve out-of-distribution robustness and overall performance (Wortsman et al., 2022). Task vectors implement additive editing and compositionality by linear arithmetic on weight differences, enabling positive and negative edits as well as multi-task synthesis (Ilharco et al., 2023). TIES-Merging explicitly addresses interfering factors such as resetting parameters with negligible updates and resolving sign conflicts, which mitigates performance degradation caused by parameter-level interference when merging multiple models (Yadav et al., 2023). Community tools and practice are also maturing. MergeKit consolidates and engineers diverse merging algorithms, supporting large-model merging and recipe reproduction under resource constraints, which facilitates methodological comparison and reproducibility (Goddard et al., 2024). Systematic surveys have begun to organize theoretical perspectives, method taxonomies, and application boundaries for merging, providing references for unified terminology, evaluation settings, and future research agendas (Yang et al., 2024b). However, most existing methods focus on average multi-task performance and out-of-distribution robustness, with limited attention to the fidelity of fine-grained functional structures such as the reasoning format, for example, explicit thought traces and process markers. RAIN-Merging targets this gap: during parameter fusion it introduces subspace constraints tied to the "thinking format," and allocates merging coefficients at the per-layer and per-head levels using instruction attention, thereby strengthening instruction following while suppressing structural drift of the original reasoning patterns.

**Null Space Projection.** Constraint ideas centered on orthogonality and null spaces have been repeatedly validated in continual learning and knowledge editing. OGD projects gradients for new tasks onto the orthogonal complement of the subspace of old tasks, explicitly constraining update

directions to mitigate forgetting (Farajtabar et al., 2020). GPM extracts and maintains "important gradient subspaces" via singular value decomposition, then performs layer-wise orthogonal projection of new gradients to reduce interference across tasks (Saha et al., 2021). For LLM knowledge editing, AlphaEdit projects edit perturbations into the null space of "preserved knowledge" and provides theoretical guarantees on output preservation, which markedly reduces cumulative damage in sequential edits (Fang et al., 2025). In parameter-efficient and mergeable settings, LoRA-Null initializes or constrains the LoRA adaptation subspace using the null space of pretrained representations, alleviating forgetting and improving parallelism and mergeability with other updates (Tang et al., 2025). For multi-task and multi-LoRA model merging, OSRM imposes orthogonalization constraints on task-specific LoRA subspaces before fine-tuning, reducing mutual interference at merge time and improving compatibility (Zhang & Zhou, 2025). Following this line of work, we construct a null-space projection on features tied to the "reasoning format," and combine it with instruction-attention–guided coefficients. The merged model thus preserves structured reasoning outputs while improving adherence to verifiable constraints such as format, length, and enumeration.

# E    PROOF

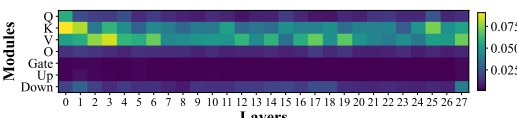

Figure A1: *Principal subspace cosine similarity* between DeepSeek-R1-Distill-Qwen-1.5B (LRM) and Qwen2.5-1.5B-Instruct (ITM) task vectors for each layer and submodule.

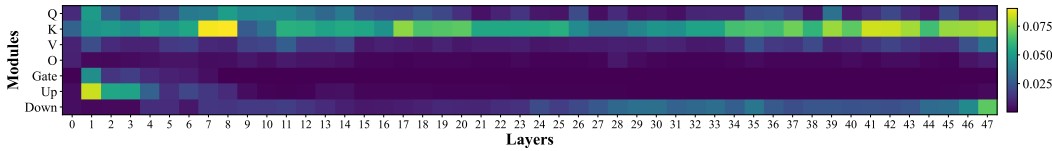

Figure A2: *Principal subspace cosine similarity* between DeepSeek-R1-Distill-Qwen-14B (LRM) and Qwen2.5-14B-Instruct (ITM) task vectors for each layer and submodule.

## E.1    PROOF OF WHY ORTHOGONAL IN PARAMETERS $\neq$ INVARIANT IN OUTPUTS

We first describe how we compute orthogonality between **principal parameter subspaces**. Let the two sources be the LRM task vector or weight difference, denoted by $R$, and the ITM counterpart, denoted by $I$. For each layer and each linear submodule $W^k \in \mathbb{R}^{d_{\text{out}}^k \times d_{\text{in}}^k}$, we take the top $S$ singular directions (default $S = 16$ in our experiments) and perform SVD:

$$W_R^k \;=\; U_R^k \, \Sigma_R^k \, (V_R^k)^\top, \qquad W_I^k \;=\; U_I^k \, \Sigma_I^k \, (V_I^k)^\top. \tag{A1}$$

Write $U_{R,S}^k \in \mathbb{R}^{d_{\text{out}}^k \times S}$ for the top-$S$ left singular vectors of $U_R^k$ (similarly $V_{R,S}^k \in \mathbb{R}^{d_{\text{in}}^k \times S}$), and analogously $U_{I,S}^k, V_{I,S}^k$ for source $I$.

**Principal subspace cosine similarity.** We focus on the *left* (output-side) principal subspaces and define the alignment matrix

$$A^k \;=\; (U_{R,S}^k)^\top U_{I,S}^k \;\in\; \mathbb{R}^{S \times S}. \tag{A2}$$

Let $\sigma_1^k, \ldots, \sigma_S^k \in [0, 1]$ be the singular values of $A^k$. They equal the cosines of the principal angles between the two subspaces: $\sigma_i^k = \cos \vartheta_i^k$. We define the **principal subspace cosine similarity** as the mean cosine of principal angles:

$$\cos \Theta_S^k \big( U_{R,S}^k, U_{I,S}^k \big) \;=\; \frac{1}{S} \sum_{i=1}^{S} \sigma_i^k. \tag{A3}$$

Smaller values indicate stronger orthogonality between the sources at that (layer, module) cell.

**Empirical observation.** Across model sizes and all layers/modules in Qwen2.5-1.5B/7B/14B (**Fig. 2**, **A1**, **A2**), we observe $\cos \Theta_S^k < 0.1$ (with only a few exceptions), indicating that LRM and ITM task vectors are largely *orthogonal in parameter principal directions*. However, as the theory below shows, such parameter-space orthogonality does *not* imply invariance in the output space (i.e., unchanged logits on the thinking format), and thus cannot replace the *null-space projection* constraint used in our method.

**Why orthogonal in parameters $\neq$ invariant in outputs.** We formalize this issue as the following **Prop. 1** and give a proof with dimension argument.

**Proposition 1** (Insufficiency of parameter-space orthogonality for output invariance)**.** *For each submodule $k$, let $\mathcal{U}_{I,S}^k, \mathcal{V}_{I,S}^k$ be the $S$-dimensional instruction-side principal left/right subspaces and define the admissible low-rank parameter perturbation space*

$$\mathcal{T}_I^k \; = \; \mathcal{U}_{I,S}^k \otimes \mathcal{V}_{I,S}^k \; = \; \text{span}\big\{\text{vec}(uv^\top) : u \in \mathcal{U}_{I,S}^k, \; v \in \mathcal{V}_{I,S}^k\big\}, \qquad \mathcal{T}_I \; = \; \bigoplus_{k=1}^K \mathcal{T}_I^k. \quad \text{(A4)}$$

*Let $J$ be the Jacobian of the logits on the protected thinking tokens $\Omega_{think}$ at the anchor $\theta_R$, with total parameter dimension $D = \dim(\theta)$ and rank $r = \text{rank}(J)$. Then, in generic position,*

$$\dim\big(\mathcal{T}_I \cap \text{Null}(J)\big) \; \leq \; \max\{0, \; KS^2 - r\}. \quad \text{(A5)}$$

*In particular, if $r > KS^2$, one has $\mathcal{T}_I \cap \text{Null}(J) = \{0\}$ and hence $\mathcal{T}_I \nsubseteq \text{Null}(J)$. Even when $r \leq KS^2$, the inclusion $\mathcal{T}_I \subseteq \text{Null}(J)$ requires a measure-zero alignment and thus almost never holds. Consequently, there exists a nonzero $\Delta \in \mathcal{T}_I$ with $J\Delta \neq 0$, implying*

$$\mathcal{L}_{think}(\theta_R + \Delta) \; = \; \tfrac{1}{2} \Delta^\top \big(\mathbb{E}[J^\top FJ]\big)\Delta \; + \; o(\|\Delta\|^2) \; > \; 0, \quad \text{(A6)}$$

*where $F$ is the Fisher matrix of the softmax.*

*Proof of Prop. 1.* Each module contributes an $S$-dimensional left subspace and an $S$-dimensional right subspace; their Kronecker product yields

$$\dim \mathcal{T}_I^k \; = \; S \cdot S \; = \; S^2 \quad \text{in generic position,} \quad \text{(A7)}$$

so, ignoring accidental cross-module dependencies,

$$\dim \mathcal{T}_I \; = \; \sum_{k=1}^K \dim \mathcal{T}_I^k \; = \; KS^2. \quad \text{(A8)}$$

By the rank–nullity theorem for $J \in \mathbb{R}^{m \times D}$,

$$\dim \text{Null}(J) \; = \; D - r. \quad \text{(A9)}$$

For two subspaces $\mathcal{A}, \mathcal{B} \subset \mathbb{R}^D$, a standard upper bound on the intersection dimension states

$$\dim(\mathcal{A} \cap \mathcal{B}) \; \leq \; \max\{0, \; \dim \mathcal{A} + \dim \mathcal{B} - D\}. \quad \text{(A10)}$$

Setting $\mathcal{A} = \mathcal{T}_I$ and $\mathcal{B} = \text{Null}(J)$ gives

$$\dim\big(\mathcal{T}_I \cap \text{Null}(J)\big) \; \leq \; \max\big\{0, \; KS^2 + (D - r) - D\big\} \; = \; \max\{0, \; KS^2 - r\}. \quad \text{(A11)}$$

Hence, if $r > KS^2$, the intersection is trivial and $\mathcal{T}_I \subseteq \text{Null}(J)$ is impossible. Even when $r \leq KS^2$, the full inclusion would require not only $\dim \mathcal{T}_I \leq \dim \text{Null}(J)$ but also a non-generic containment (measure-zero alignment) between the two subspaces; thus it almost never holds in generic position.

Finally, since $F \succeq 0$ and $M = \mathbb{E}[J^\top FJ] \succeq 0$, any nonzero $\Delta \in \mathcal{T}_I$ with $J\Delta \neq 0$ satisfies $\Delta^\top M \Delta > 0$, yielding

$$\mathcal{L}_{\text{think}}(\theta_R + \Delta) \; = \; \tfrac{1}{2} \Delta^\top M \Delta + o(\|\Delta\|^2) \; > \; 0. \quad \text{(A12)}$$

$\square$

In words, *orthogonality of principal parameter subspaces* does *not* guarantee *first-order invariance of outputs* on the thinking format. This is precisely why our Stage 1 imposes a *null-space projection* constraint (i.e., $\Phi_{\Omega_{\text{think}}} \text{vec}(\Delta^\perp) = 0$) to cancel first-order effects.

E.2   PROOF OF PROP. 1

*Proof of Prop. 1.* Let $p = \text{softmax}(z) \in \Delta^{V-1}$ and $q = \text{softmax}(z + u)$, where $z \in \mathbb{R}^V$ is the logits vector at a thinking position $t \in \Omega_{\text{think}}(x)$ and $u \in \mathbb{R}^V$ is the perturbation induced by the parameter change.

**Step 1: KL as a Bregman divergence of** lse **and a uniform quadratic bound.** Let $\text{lse}(z) = \log\sum_{i=1}^{V} e^{z_i}$, so that $\nabla \text{lse}(z) = \text{softmax}(z) = p$ and $\nabla^2 \text{lse}(z) = \text{diag}(p) - pp^\top$. For the multinomial exponential family, the KL divergence equals the Bregman divergence of the log-partition function (Banerjee et al., 2005; Wainwright & Jordan, 2008):

$$\text{KL}(\text{softmax}(z + u) \,\|\, \text{softmax}(z)) = D_{\text{lse}}(z+u, z) = \text{lse}(z+u) - \text{lse}(z) - \langle \nabla \text{lse}(z), u \rangle. \quad \text{(A13)}$$

Using the integral form of the Bregman remainder for a twice differentiable convex $f$, $D_f(x+h, x) = \int_0^1 (1-s)\, h^\top \nabla^2 f(x+sh)\, h\, ds$, and the fact that for all $z$ the Hessian satisfies $\|\nabla^2 \text{lse}(z)\|_2 \leq \frac{1}{4}$ by positive semidefinite covariance form as in Lemma 1 (Boyd & Vandenberghe, 2004; Böhning, 1992), we obtain

$$\text{KL}(\text{softmax}(z + u) \,\|\, \text{softmax}(z)) = \int_0^1 (1-s)\, u^\top \nabla^2 \text{lse}(z + su)\, u\, ds$$

$$\leq \int_0^1 (1-s)\, \tfrac{1}{4}\|u\|_2^2\, ds = \tfrac{1}{8}\|u\|_2^2. \quad \text{(A14)}$$

Equivalently, the second-order Taylor expansion with a third-order remainder yields

$$\text{KL}(\text{softmax}(z + u) \,\|\, \text{softmax}(z)) = \tfrac{1}{2}\, u^\top \nabla^2 \text{lse}(z)\, u + O(\|u\|_2^3) \leq \tfrac{1}{8}\|u\|_2^2 + O(\|u\|_2^3). \quad \text{(A15)}$$

**Lemma 1** (Hessian bound for lse). *For any $z \in \mathbb{R}^V$ with $p = \text{softmax}(z)$,*

$$\nabla^2 \text{lse}(z) = \text{diag}(p) - pp^\top \succeq 0, \qquad \left\|\nabla^2 \text{lse}(z)\right\|_2 \leq \tfrac{1}{4}. \quad \text{(A16)}$$

**Step 2: Bounding the logits perturbation via linearization and Lipschitz regularity.** Let $J(x,t) \in \mathbb{R}^{V \times d}$ be the Jacobian of $z_\theta(x,t)$ w.r.t. $\theta$ at $\theta = \theta_R$. By the mean-value theorem and Taylor expansion with Lipschitz gradient (Nesterov, 2013),

$$u(x,t) := z_{\theta_R + \Delta}(x,t) - z_{\theta_R}(x,t) = J(x,t)\,\text{vec}(\Delta) + r(x,t), \quad \|r(x,t)\|_2 \leq \tfrac{L}{2}\,\|\Delta\|_2^2, \quad \text{(A17)}$$

where $L$ is a local Lipschitz constant of $\nabla_\theta z_\theta(x,t)$ around $\theta_R$. Let $\Phi = \text{blkdiag}(\Phi^1, \ldots, \Phi^K)$ be the block-diagonal *forward feature operator* that maps $\text{vec}(\Delta)$ to the linearized token-level feature change collected at thinking positions (per submodule $k$). Under bounded intermediate activations and operator norms, which are standard in local linearization of deep nets (Fazlyab et al., 2019), there exists $C_1 > 0$ such that $\|J(x,t)\,\text{vec}(\Delta)\|_2 \leq C_1\,\|\Phi\,\text{vec}(\Delta)\|_2$. Combining with **Eq. (A17)**,

$$\|u(x,t)\|_2 \leq C_1\,\|\Phi\,\text{vec}(\Delta)\|_2 + C_2\,\|\Delta\|_2^2, \qquad C_2 := \tfrac{L}{2}. \quad \text{(A18)}$$

**Step 3: Enforcing the null-space constraint and aggregating into $\mathcal{L}_{\text{think}}$.** Apply the submodule-wise null-space projection (see **Eq. (5)** in the main text):

$$\text{vec}(\Delta_I^{\perp,k}) = P^\perp\big(\Phi_{\Omega_{\text{think}}}^k\big)\,\text{vec}(\Delta_I^k), \qquad \Delta_I^\perp = \bigoplus_{k=1}^{K} \Delta_I^{\perp,k}, \quad \text{(A19)}$$

so that by construction $\Phi_{\Omega_{\text{think}}}\,\text{vec}(\Delta_I^\perp) = 0$. Plugging this into **Eq. (A18)** yields for all $t \in \Omega_{\text{think}}(x)$:

$$\|u(x,t)\|_2 \leq C_2\,\|\Delta_I^\perp\|_2^2. \quad \text{(A20)}$$

Combining with **Eq. (A14)** and summing/averaging over $(x,t)$ in the definition of $\mathcal{L}_{\text{think}}$ (**Eq. (2)**) gives

$$\mathcal{L}_{\text{think}}\big(\theta_R + \Delta_I^\perp\big) = \mathbb{E}_x \sum_{t \in \Omega_{\text{think}}(x)} \text{KL}\Big(\pi_{\theta_R + \Delta_I^\perp}(\cdot \mid x, y_{<t}) \,\|\, \pi_{\theta_R}(\cdot \mid x, y_{<t})\Big)$$

$$\leq \tfrac{1}{8}\,\mathbb{E}_{x,t}\big[\|u(x,t)\|_2^2\big] + O\big(\mathbb{E}_{x,t}\|u(x,t)\|_2^3\big)$$

$$= O\big(\|\Delta_I^\perp\|_2^4\big) \leq O\big(\|\Delta_I^\perp\|_2^2\big) \approx 0. \quad \text{(A21)}$$

This completes the proof. □

---

**Algorithm 1:** RAIN-Merging: Reasoning-Aware Instruction-attention guided Null-space projection Merging

---

**Input** : LRM $\theta_R$; ITM $\theta_I$; base model $\theta_B$; reasoning calibration set $\mathcal{D}_R$ with thinking indices $\Omega_{\text{think}}$; instruction calibration set $\mathcal{D}_I$ with spans $(\mathcal{I}, \mathcal{R}, \mathcal{U})$; hyperparameters $\rho$, $\tilde{\alpha}_l$, $\tilde{\alpha}_u$, $\lambda$.

**Output** : Merged model $\theta_\star$.

---

1   **Stage 0: Task vector and objective.**

2   $\Delta_I \leftarrow \theta_I - \theta_B$             `// instruction-tuned task vector`

3   **Stage 1: Reasoning-aware Null-space Projection (satisfy Eq. (2)).**

4   **for** $k \leftarrow 1$ **to** $K$   `// iterate over submodules (per-layer` $W_Q, W_K, W_V, W_O,$ `FFN)` **do**

5      $\Phi_\Omega^k \leftarrow$ FeatureOperator $(\theta_R, \mathcal{D}_R, \Omega_{\text{think}}, k)$   `// forward feature extraction at thinking tokens`

6      $P_k^\perp \leftarrow \text{diag}(1) - (\Phi_\Omega^k)^\top (\Phi_\Omega^k (\Phi_\Omega^k)^\top + \text{diag}(1))^{-1} \Phi_\Omega^k$    `// least-squares orthogonal projector`

7      $\text{vec}(\Delta_I^{\perp,k}) \leftarrow P_k^\perp \text{vec}(\Delta_I^k)$          `// submodule projection per` **Eq. (5)**

8   $\theta' \leftarrow \theta_R + \bigoplus_{k=1}^K \Delta_I^{\perp,k}$          `// direct merge after Stage 1`

9   **Stage 2: Instruction-attention Guided Merging Coefficients (optimize Eq. (11)).**

10   Initialize head-wise coefficients $\tilde{\alpha}^{\tilde{k}} \leftarrow 1$ for all attention heads $\tilde{k}$.

11   **for** *each attention head* $\tilde{k}$ **do**

12      $a^{\tilde{k}} \leftarrow \mathbb{E}_{x \sim \mathcal{D}_I}\Big[ \frac{1}{|\mathcal{I}(x)||\mathcal{R}(x)|} \sum_{t \in \mathcal{R}(x)} \sum_{\tau \in \mathcal{I}(x)} \text{Att}_{\theta'}^{\tilde{k}}(x)[t,\tau]\Big]$

13      $u^{\tilde{k}} \leftarrow \mathbb{E}_{x \sim \mathcal{D}_I}\Big[ \frac{1}{|\mathcal{I}(x)||\mathcal{U}(x)|} \sum_{t \in \mathcal{U}(x)} \sum_{\tau \in \mathcal{I}(x)} \text{Att}_{\theta'}^{\tilde{k}}(x)[t,\tau]\Big]$

14   **for** *each attention head* $\tilde{k}$ **do**

15      $g^{\tilde{k}} \leftarrow a^{\tilde{k}} - \rho\, u^{\tilde{k}}$          `// first-order term for` **Eq. (11)**

16      $\tilde{H}^{\tilde{k}} \leftarrow 1 + u^{\tilde{k}}$          `// diagonal Hessian approx`

17      $\tilde{\alpha}_\star^{\tilde{k}} \leftarrow \text{clip}_{[\tilde{\alpha}_l, \tilde{\alpha}_u]}\Big(\frac{g^{\tilde{k}}}{\tilde{H}^{\tilde{k}}}\Big)$          `// per-head optimal scaling`

18   $\alpha_\star^k \leftarrow$ Aggregate $\big(\{\tilde{\alpha}_\star^{\tilde{k}}\}_{\tilde{k} \in \text{module } k}\big)$          `// mean over heads for FFN`

19   **Output (Two-stage Merge).**

20   **return** $\theta_\star \leftarrow \theta_R + \lambda \bigoplus_{k=1}^K \alpha_\star^k \Delta_I^{\perp,k}$          `// final model in` **Eq. (16)**

---

# F   ALGORITHM

Following **Alg.** 1 is the algorithm of our RAIN-Merging.

# G   METHOD IMPLEMENTATION DETAILS

## G.1   FORWARD MECHANISM IN TRANSFORMER

A standard Transformer layer consists of multi-head self-attention and a feed-forward network (FFN). In layer $\ell$, the hidden state of the token at position $t$, denoted $h_t^{(\ell-1)} \in \mathbb{R}^d$, is linearly projected to queries, keys, and values: $q_t^{(\ell)} = W_Q^{(\ell)} h_t^{(\ell-1)}, k_\tau^{(\ell)} = W_K^{(\ell)} h_\tau^{(\ell-1)}, v_\tau^{(\ell)} = W_V^{(\ell)} h_\tau^{(\ell-1)}$. For head $h$, the single-head attention weights are $\text{Att}_\theta^{(\ell,h)}(x)[t,\tau] = \text{softmax}_\tau\big(\langle q_t^{(\ell,h)}, k_\tau^{(\ell,h)}\rangle / \sqrt{d_k}\big)$, which represent the probability that the token at position $t$ attends to position $\tau$. The corresponding head output is $o_t^{(\ell,h)} = \sum_\tau \text{Att}_\theta^{(\ell,h)}(x)[t,\tau]\, v_\tau^{(\ell,h)}$. After concatenating the outputs from all heads and applying $W_O^{(\ell)}$, we obtain $\tilde{h}_t^{(\ell)}$. The FFN then computes $\hat{h}_t^{(\ell)} = \sigma\big(W_{\text{in}}^{(\ell)} \tilde{h}_t^{(\ell)} + b_{\text{in}}^{(\ell)}\big), h_t^{(\ell)} = W_{\text{out}}^{(\ell)} \hat{h}_t^{(\ell)} + b_{\text{out}}^{(\ell)}$. The top-layer hidden state is mapped to vocabulary logits $z_\theta(x,t)$, which are transformed by a softmax into the conditional distribution $\pi_\theta(\cdot \mid x, y_{<t})$. We follow the notation and the scaled dot-product attention definition of Vaswani et al. (2017) to align with prior work.

Table A1: Reasoning calibration set construction from *Mixture-of-Thoughts*. We uniformly sample 50 examples per domain for calibration and 50 for validation. Raw sizes are taken from the official dataset composition page.

| Domain | Raw size | Calibration | Validation |
|---|---|---|---|
| Math | 93,700 | 50 | 50 |
| Code | 83,100 | 50 | 50 |
| Science | 173,000 | 50 | 50 |
| **Total** | 349,800 | 150 | 150 |

## G.2 IMPLEMENTATION DETAILS IN MERGING

To balance computational efficiency and memory usage, all model-merging experiments adopt a **layer-wise** merging strategy. During parameter fusion, we compute in **FP64** precision to ensure numerical stability, and we store the final models in **BF16**. Our experiments show that higher compute precision yields consistent but modest improvements for this merging procedure.

## H CALIBRATION SET CONSTRUCTION

### H.1 REASONING CALIBRATION SET

We employ the *Mixture-of-Thoughts*[2] (Face, 2025) dataset as the source for reasoning-style calibration. This dataset contains validated R1-style reasoning traces spanning three domains: math, code, and science, with a total size of approximately 350k samples. Its official data composition page clearly specifies the sample sizes and origins for each sub-domain: math samples are sourced from OpenR1-Math (Lozhkov et al., 2025), code from CodeForces-CoTs (Penedo et al., 2025), and science from the science subset of the Nemotron post-training set (Bercovich et al., 2025). From each domain, we randomly sample 50 instances to form the calibration set (150 in total), and an additional 50 instances per domain are randomly sampled to serve as the validation set (150 in total). **Tab.** A1 shows the detailed numbers of samples in each domain.

**Thinking Special Token Set Construction.** To apply preservation constraints on "thinking format", we extract the thinking tokens, specifically `<think>` and `</think>` in the model output—based on the R1-style chat template and tokenizer. The procedure involves rendering messages using the chat template provided by LRM. R1-family models prefill `<think>` in reasoning mode and insert `</think>` in the context, while some templates may omit the visible output of the initial `<think>` to enforce thinking mode. We then obtain token positions of `<think>` and `</think>` in $\Omega_{\text{think}}$.

### H.2 INSTRUCTION CALIBRATION SET

We construct a high-quality instruction calibration set from rule-verifiable prompts through four automated and auditable steps. The pipeline produces span-based samples $(x \sim \mathcal{D}_I; \mathcal{I}(x), \mathcal{R}(x), \mathcal{U}(x))$ for computing the instruction-attention score proxy in Stage 2 of RAIN-Merging. We choose to distill from IFEval-style instructions for ease of implementation and to test generalization on out-of-domain instruction-following datasets. The final size of the instruction calibration set is 365. The full workflow is:

- **Instruction selection.** We select rule-verifiable instruction prompts from IFEval (Zhou et al., 2023b) as queries. Each record contains a natural-language instruction and machine-checkable constraints.
- **Step 1: Response generation by LRM.** For each instruction query, we invoke an R1-style reasoning model (`deepseek-reasoner`, DeepSeek-R1-0528)[3] to produce a format-explicit response. This step yields instruction-following samples generated by a reasoning decoder that reflect realistic decoding behavior.

---

[2]https://huggingface.co/datasets/open-r1/Mixture-of-Thoughts
[3]https://api.deepseek.com/v1

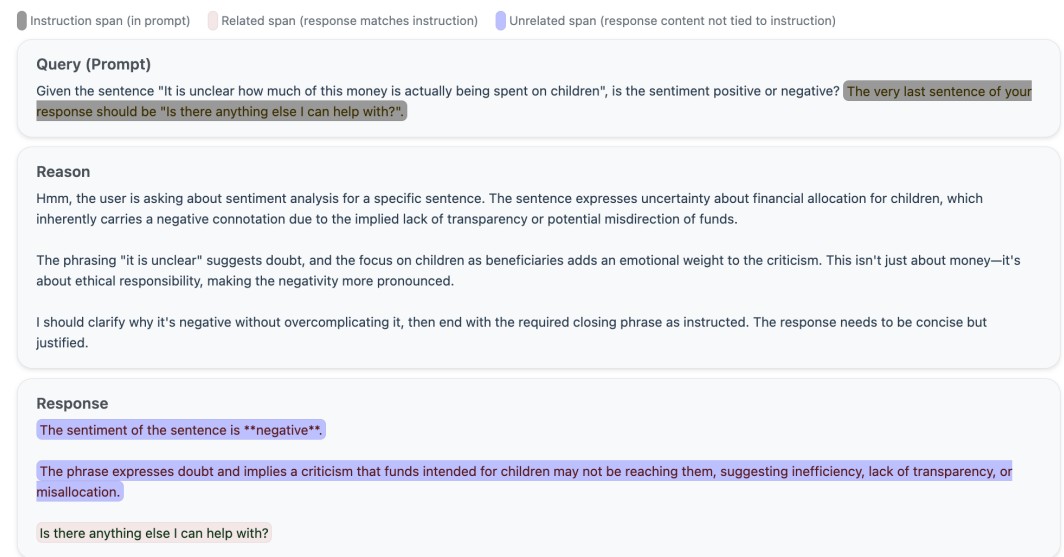

Figure A3: A sample illustration in our instruction calibration set.

Table A2: Instruction-following benchmarks. We list dataset size, constraint taxonomy, composition types, verification, and aggregation strategy.

| Benchmark | Size | Constraint Taxonomy | Composition Type | | | | Verification | | Evaluation |
|---|---|---|---|---|---|---|---|---|---|
| | | | And | Chain | Selection | Nested | Code-Exec. | LLM-as-Judge | Aggregation |
| IFEval | 541 | 25 | ✓ | – | – | – | ✓ | – | strict_prompt_level_accuracy |
| CELLO | 523 | 4 | ✓ | ✓ | – | – | ✓ | – | average |
| InfoBench | 500 | 5 | ✓ | ✓ | – | – | – | ✓ | DRFR |
| ComplexBench | 1,150 | 19 | ✓ | ✓ | ✓ | ✓ | ✓ | ✓ | dependency-aware DRFR |

- **Step 2: Rule evaluation and filtering.** We evaluate the outputs of Step 1 with an IFEval-compatible checker and *retain only passing samples* that satisfy all constraints. This removes cases that clearly fail the requirements.

- **Step 3: Strict span extraction (LLM-as-Judge).** We use a high-performance instruction-tuned LLM (`deepseek-chat`, DeepSeek-V3.1)[3] to precisely extract instruction-relevant spans: $\mathcal{I}(x)$ (tokens in the prompt corresponding to the instruction) and $\mathcal{R}(x)$ (tokens in the response that are governed by the instruction). The unrelated span $\mathcal{U}(x)$ is then implicitly defined as the remainder of the response. See **Fig. A3** for an example.

- **Step 4: Tokenizer-level verification.** We verify the extracted spans on the *target tokenizer* (aligned with our anchor LRM), ensuring that boundaries lie on token edges and can be deterministically reconstructed. Samples that fail alignment are discarded.

- **Step 5: Human review and ethical screening.** To ensure data quality and compliance with safety and ethics standards, we introduce a manual review stage. Researchers verify the accuracy of the LLM-extracted spans $\mathcal{I}(x)$ and $\mathcal{R}(x)$, and conduct an ethics audit of the responses based on content-safety guidelines, removing any samples that contain biased, harmful, or inappropriate content. This step further enhances the reliability and ethical soundness of the calibration set.

This calibration pipeline is readily transferable and can be extended to additional instruction-following datasets to further improve merging effectiveness by enriching the calibration set. The reasoning distillation model and the LLM-as-Judge can be updated over time to continually enhance the quality of the instruction calibration data.

Table A3: Test set sizes of the six math benchmarks used in our mathematical reasoning (Math) evaluation.

|  | AIME2025 | AIME2024 | AMC23 | GSM8K | Math500 | MinervaMath |
|---|---|---|---|---|---|---|
| # Test samples | 30 | 30 | 40 | 1,319 | 500 | 272 |

# I    DETAILED EXPERIMENTAL SETUP

## I.1    BENCHMARKS

**Instruction-following Benchmarks.**    We evaluate instruction compliance on four widely used, programmatically verifiable benchmarks. The size and constraint types are summarized in **Tab.** A2.

- **IFEval** (Zhou et al., 2023b). IFEval provides four accuracy metrics: (1) *prompt-level strict* accuracy and (2) *instruction-level strict* accuracy, plus (3) *prompt-level loose* and (4) *instruction-level loose* variants. The strict metrics require exact satisfaction (all constraints per prompt for prompt-level strict; per-constraint averaging across prompts for instruction-level strict). The loose metrics first normalize model outputs (e.g., strip Markdown, boilerplate intros/outros) to reduce false negatives. We report the official **strict_prompt_level_accuracy** unless otherwise noted.

- **CELLO** (He et al., 2024). CELLO uses a *code-based* verifier that scores four granular aspects: (i) *count limit* (word/sentence/sample counts), (ii) *answer format* (parsability, keywords), (iii) *task-prescribed phrases* (mandatory phrases covered), and (iv) *input-dependent query* (presence of key phrases from the input), with a penalty to discourage verbatim copying. We follow the benchmark's practice and **average** these checks to produce the final score.

- **InfoBench** (Qin et al., 2024). InfoBench adopts the *Decomposed Requirements Following Ratio (DRFR)*: each instruction is split into scoring questions that are judged by *LLM-as-a-Judge* with binary YES/NO labels; the final score is the mean over all questions, enabling fine-grained interpretability. We evaluate by GPT-5-mini. We report the official **DRFR**.

- **ComplexBench** (Wen et al., 2024). ComplexBench also evaluates via decomposed scoring questions with *YES/NO* judgments and aggregates them into DRFR, but crucially uses a *dependency-aware* scheme: if any prerequisite constraint fails, all dependent (downstream) constraints are automatically marked as failed. This better reflects multi-constraint compositions. We evaluate by GPT-4o-mini. We report the **dependency-aware DRFR**.

**Reasoning & General Benchmarks.**    We evaluate reasoning and general capabilities on the following benchmarks:

- **Math (Mathematical reasoning).** We aggregate **Pass@1/accuracy** over six common math benchmarks: **AIME2025** (Balunović et al., 2025); **AIME2024** (Balunović et al., 2025); **AMC23** (Cao et al., 2024); **GSM8K** (Cobbe et al., 2021); **Math500** (Hendrycks et al., 2021); **MinervaMath** (Lewkowycz et al., 2022). The size of each math benchmark is shown in **Tab.** A3. We report the **averaged accuracy** over all benchmarks.

- **Aider** (Aider, 2024) **(Code editing).** Aider-Edit assesses code editing capability under a minimal-edit paradigm. It contains 133 small Python coding exercises sourced from Exercism, where the model is provided with a natural-language edit instruction and the existing code, and must generate a correct patch. The generated patch is required to apply successfully to the codebase and pass compilation and associated tests. Performance is measured by the **Pass@2 Edit Success Rate**.

- **GPQA** (Rein et al., 2024) **(STEM).** A curated, expert-level subset of GPQA comprising 198 four-option multiple-choice questions across biology, chemistry, and physics. Items are selected to be "Google-proof" and unambiguous: both expert validators must answer correctly while at most one of three skilled non-experts succeeds, yielding a particularly hard split. We follow common practice and report **accuracy** (strict single-choice).

- **Arena-Hard-v2** (Li et al., 2024) **(Creative writing).** A hard, open-ended benchmark constructed to maximize model separability and align with human preferences. Arena-Hard-Auto curate ~500 challenging prompts covering difficult real-world tasks including creative writing, scoring follows the pairwise battle paradigm with human or LLM-as-a-Judge assessments, and results are commonly summarized as *win rate* or transformed to *Elo* scores. For reproducibility, we fix

`Qwen2.5-7B` as the reference baseline and use `GPT-5-mini` as the judging LLM; we report the resulting **win rate** of each model against this baseline on the official Arena-Hard-v2 prompt set.

- **Agentic Scenarios.** We evaluate the ability of the model to interact with the environment and complete tasks by two agentic benchmarks: **ALFWorld** (Shridhar et al., 2021) is a text-interactive household-task environment involving multi-step planning and execution. We evaluate on 100 tasks. Metrics are **Success Rate** of goal completion. **WebShop** (Yao et al., 2022) is a web shopping agent task (search, click, compare), with reported **Normalized Reward** to capture both path efficiency and goal matching. We evaluate on 100 tasks.

## I.2 BASELINES

We compare our method with the following merging baselines:

- **Task Arithmetic** (Ilharco et al., 2023). The simplest linear composition injects the task vector additively near the anchor, $\theta = \theta_R + \lambda \Delta_I$. A scalar $\lambda \in [0, 1]$ usually controls the strength; using the same $\lambda$ per layer or per block is also common. Its advantages are zero data and negligible compute; its drawback is that conflicts across submodules are hard to disentangle.

- **SLERP** (Biship, 2007; Goddard et al., 2024). In SLERP (*Spherical Linear Interpolation*), weights are $\ell_2$-normalized to the unit sphere and interpolated along the geodesic to preserve norm and angular geometry. Let $\Omega = \arccos(\langle w_R, w_I \rangle)$. Then $\text{slerp}(w_R, w_I; t) = \frac{\sin((1-t)\Omega)}{\sin \Omega} w_R + \frac{\sin(t\Omega)}{\sin \Omega} w_I$, $t \in [0, 1]$. During merging, we apply SLERP to each tensor and rescale by the original norm. This reduces norm drift compared with linear interpolation.

- **Karcher** (Nielsen & Bhatia, 2013; Goddard et al., 2024). On a chosen manifold like the unit sphere or a Stiefel manifold, compute the Fréchet mean by minimizing the sum of squared geodesic distances: $\min_{\bar{w}} \sum_i d^2(\bar{w}, w_i)$. The iterative update is $\bar{w}^{(t+1)} = \text{Exp}_{\bar{w}^{(t)}} \left( \frac{1}{n} \sum_i \text{Log}_{\bar{w}^{(t)}}(w_i) \right)$.

- **TIES** (Yadav et al., 2023). TIES is a data-free method that explicitly prunes and sparsifies to handle parameter-level conflicts. For each layer's edit vector it applies (i) a *sign-consistency mask* (retain entries aligned with the dominant direction to reduce cancellation), (ii) *magnitude thresholding or Top-$k$ truncation* (keep high-contribution entries and zero out the rest), and (iii) optional *rescaling* to match a target norm. We can stack TIES as a post-processing step on top of feasible baselines to improve robustness.

- **DARE** (Yu et al., 2024). DARE uses first-order sensitivities on a small calibration set (for example, gradient norms of labeled loss, log-likelihood changes, or Fisher approximations of the output distribution) to learn a per-layer or per-tensor coefficient $\alpha^k$ (or a diagonal preconditioner), yielding $\theta = \theta_R + \bigoplus_k \alpha^k \Delta_I^k$. It can be viewed as data-aware recalibration that reduces the bias introduced by naive addition with very low compute.

- **ACM** (Yao et al., 2025). ACM (*Activation-Guided Consensus Merging*) targets activation consistency. On a small calibration set it measures, before and after injecting the task vector, how each layer or head changes its response on instruction-relevant spans and its leakage on irrelevant spans. It then solves for per-submodule coefficients $\alpha^k$, optionally with cross-sample consensus regularization to improve generalization.

- **LEWIS** (Chopra et al., 2025). LEWIS (*LayEr WIse Sparsity*) is a merge with layer-wise sparsity allocation. Based on sensitivity indicators per layer (such as edit-vector magnitude, activation gradients, or Fisher approximations), it sets a budget $s_k$ and merges only the Top-$s_k$ parameters of that layer, leaving the anchor weights elsewhere unchanged. It can be combined with Task Arithmetic, SLERP, or Karcher as the base, and $\{s_k\}$ are determined by heuristics or grid search on a small calibration set.

- **AIM** (Nobari et al., 2025). AIM (*Activation-based Importance Merging*) weights the task vector by activation importance (for example, the effect of Value or FFN outputs on downstream logits, or the instruction-aligned component of attention weights), performing element-wise or block-wise reweighting: Weights the task vector by activation importance, for example, the effect of Value or FFN outputs on downstream logits, or the instruction-aligned component of attention weights, performing element-wise or block-wise reweighting: $\theta = \theta_R + \bigoplus_k W^k \odot \Delta_I^k$, where $W^k$ is obtained from a single forward pass on the calibration set. Intuitively, this preserves edits that meaningfully change useful representations and suppresses noisy updates.

Table A4: The hyperparameters of various merging methods in **Tab.** 1. $\lambda$ means the global scaling coefficient in merging. $k$ denotes the trim ratio in TIES-Merging. $p$ means the drop rate in DARE merging. $\tau$ is sharpness the ACM. $\rho$ is the pruning ratio in LEWIS. $\omega$ means the balance factor in AIM.

| Method | Hyper-parameters |
|---|---|
| | **7B** |
| Task Arithmetic | $\lambda = 1.0$ |
| SLERP | $\lambda = 1.0$ |
| Karcher | $\lambda = 1.0$ |
| TIES | $k = 0.8,\ \lambda = 0.8$ |
| DARE-TIES | $p = 0.3,\ k = 0.5,\ \lambda = 1.2$ |
| ACM-TIES | $\tau = 1.0,\ k = 0.5,\ \lambda = 1.1$ |
| LEWIS-TIES | $\rho = 0.5,\ k = 0.5,\ \lambda = 1.1$ |
| AIM-TIES | $\omega = 0.4,\ k = 0.5,\ \lambda = 1.0$ |

Table A5: The hyperparameters of RAIN-Merging in different model sizes. $\lambda$ means the global scaling coefficient in RAIN-Merging.

| Method | Hyper-parameters | | | |
|---|---|---|---|---|
| | **1.5B** | **7B** | **8B** | **14B** |
| RAIN-Merging | $\lambda = 1.0$ | $\lambda = 1.0$ | $\lambda = 0.9$ | $\lambda = 1.0$ |

Unless otherwise specified, following common practice in previous work (Wu et al., 2025), we apply TIES post-processing (sign consistency and magnitude truncation) on the outputs of DARE, ACM, LEWIS, and AIM, in order to improve comparability across baselines.

### I.3 HYPERPARAMETERS

For SFT, we use a batch size of 16 with the Adam optimizer (Kingma & Ba, 2014), a learning rate of $2 \times 10^{-5}$, weight decay of 0.05, and train for 20 epochs.

For all model-merging methods (including the proposed RAIN-Merging and all baselines), we merge only the task vectors extracted from the ITM's Q/K/V/O/FFN modules. The specific hyperparameter settings for each baseline used in **Tab.** 1 are listed in **Tab.** A4.

In **RAIN-Merging**, we set the leakage penalty to $\rho = 10$ and bound the attention-head coefficients by $[\tilde{\alpha}_l, \tilde{\alpha}_u] = [0.0, 1.0]$. The global scalar $\lambda$ is selected via a grid search over $[0.0, 1.5]$ with a step size of 0.1; the chosen values for different model families are provided in **Tab.** A5. An ablation study of the global scalar $\lambda$ is included in Appendix J.2.

## J ADDITIONAL EXPERIMENTS

### J.1 DETAILED MATH BENCHMARK RESULTS

**Tab.** A6 and **Tab.** A7 report detailed results on the mathematics benchmarks. **RAIN-Merging** consistently preserves the mathematical reasoning ability of LRMs across different model sizes and architectures. In some cases, improving instruction following also correlates with better mathematical performance, suggesting that enhanced adherence can support clearer intermediate reasoning and more reliable final answers.

### J.2 ABLATION STUDY OF THE GLOBAL SCALAR

We conduct a sensitivity analysis of the global scalar $\lambda$ (**Fig.** A4). Across a wide range around our chosen value near 1.0, the merged model maintains strong instruction-following performance. As $\lambda$

Table A6: Math benchmark results under the same configuration as in **Tab.** 1. "Avg." denotes the average over all math benchmarks. The best and second-best results are highlighted in **bold** and underlined, respectively.

| Method | AIME2025 | AIME2024 | AMC23 | GSM8K | Math500 | Minerva | Avg. |
|--------|----------|----------|-------|-------|---------|---------|------|
| ITM | 10.00 | 10.00 | 67.50 | 86.66 | 73.80 | 35.66 | 47.27 |
| LRM | 30.00 | 50.00 | 80.00 | 91.36 | 89.00 | 48.16 | 64.75 |
| SFT | 33.33 | 43.33 | 75.00 | 90.75 | 87.40 | 45.59 | 62.57 |
| Task Arithmetic | 30.00 | 50.00 | 80.00 | 90.75 | 89.00 | 45.59 | 64.22 |
| SLERP | 30.00 | 46.67 | 77.50 | 91.05 | 89.20 | 48.53 | 63.82 |
| Karcher | 30.00 | 50.00 | 80.00 | 90.98 | 89.20 | 48.90 | 64.85 |
| TIES | 33.33 | 46.67 | 82.50 | 91.36 | **90.60** | 48.16 | 65.44 |
| DARE-TIES | **36.67** | 40.00 | 82.50 | 90.98 | 90.20 | 45.22 | 64.26 |
| ACM-TIES | 33.33 | 50.00 | 77.50 | 91.36 | 88.20 | 47.06 | 64.57 |
| LEWIS | 30.00 | 33.33 | 80.00 | 90.27 | 88.80 | **50.00** | 62.07 |
| AIM-TIES | 33.33 | 50.00 | **85.00** | 89.76 | 89.60 | 47.79 | 65.92 |
| RAIN-Merging | **36.67** | **60.00** | **85.00** | **92.12** | 90.20 | 48.53 | **68.75** |

Table A7: Math benchmarks results under the same configuration as in **Tab.** 2. "Avg." denotes the average over all math benchmarks.

| Method | AIME2025 | AIME2024 | AMC23 | GSM8K | Math500 | Minerva | Average |
|--------|----------|----------|-------|-------|---------|---------|---------|
| Qwen2.5-1.5B-Instruct | 3.33 | 0.00 | 30.00 | 75.44 | 59.40 | 22.43 | 31.77 |
| DeepSeek-R1-Distill-Qwen-1.5B | 20.00 | 13.33 | 42.50 | 73.77 | 71.80 | 28.31 | 41.62 |
| Qwen2.5-1.5B-RAIN-Merging | 20.00 | 16.67 | 60.00 | 76.36 | 72.40 | 29.78 | 45.87 |
| Llama-3.1-8B-Instruct | 3.33 | 6.67 | 20.00 | 81.65 | 67.60 | 34.26 | 35.59 |
| DeepSeek-R1-Distill-Llama-8B | 30.00 | 40.00 | 75.00 | 90.52 | 80.40 | 45.37 | 60.21 |
| Llama-3.1-8B-RAIN-Merging | 30.00 | 43.33 | 77.50 | 90.30 | 82.80 | 47.79 | 61.95 |
| Qwen2.5-14B-Instruct | 20.00 | 13.33 | 65.00 | 93.18 | 80.00 | 44.85 | 52.73 |
| DeepSeek-R1-Distill-Qwen-14B | 50.00 | 56.67 | 92.50 | 93.22 | 89.00 | 52.49 | 72.31 |
| Qwen2.5-14B-RAIN-Merging | 50.00 | 63.33 | 92.50 | 94.37 | 91.00 | 56.25 | 74.58 |

increases, reasoning ability improves slowly at first but then drops sharply beyond 1.0, indicating that overly large merge strength can still harm reasoning.

## J.3 ABLATION STUDY OF REASONING CALIBRATION SET SIZE

**Fig.** A5 presents an ablation over the size of the reasoning calibration set. As the set grows, preservation of reasoning improves; however, instruction-following performance degrades gradually. We hypothesize that overly strict preservation of the thinking format can limit gains in instruction adherence and also increase computation. Balancing performance and resource usage, we select a calibration size of 150.

## J.4 VISUALIZATION OF MERGING COEFFICIENTS IN STAGE 2

As shown in **Fig.** A6, the heatmap of merging coefficients for DeepSeek-R1-Distill-Qwen-7B exhibits clear layer-wise differences, indicating that different layers respond to instruction focus to different degrees. Notably, the earliest layers show the strongest response, with coefficients reaching the upper bound, and this pattern is consistent with the observations in **Fig.** 6.

## J.5 ABLATION STUDY OF INSTRUCTION CALIBRATION SET GENERALIZATION

While we use an instruction calibration set from IFEval in the main experiments due to its cleanly separable instruction spans and rule-based labels, the Stage 2 proxy is not restricted to such data. we construct an additional instruction calibration set from InfoBench. InfoBench focuses on open-ended constraints such as tone, style and content focus. We follow the same filtering pipeline as in Appendix H.2 and perform manual screening to ensure that the selected spans correspond to

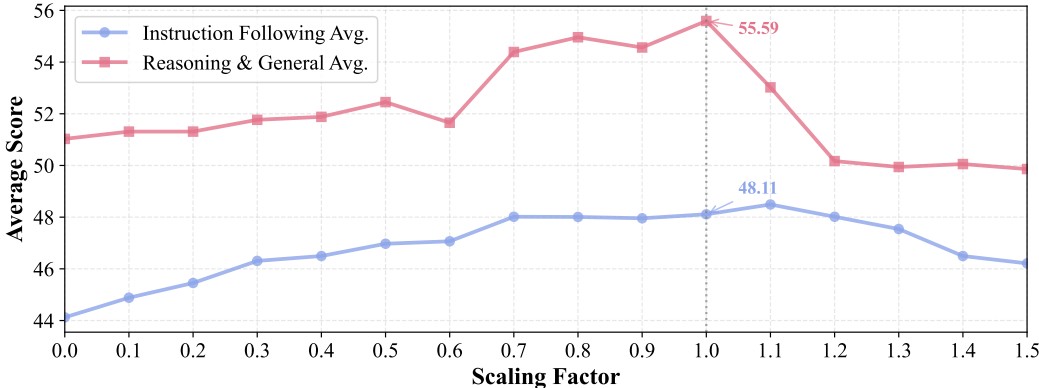

Figure A4: Instruction following and reasoning & generage performance of our RAIN-Merging using different global scalar $\lambda$. The configuration is the same as in **Tab.** 2. The performance is measured by the average of the instruction following and reasoning & general capability benchmarks. The marked result is our choice in the experiments.

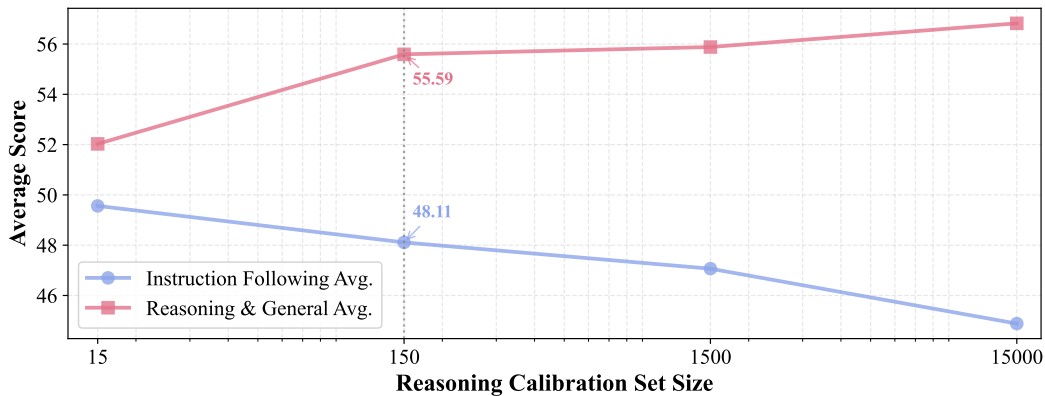

Figure A5: Instruction following and reasoning & generage performance of our RAIN-Merging using different reasoning calibration set sizes. The $x$ axis represents the size of the reasoning calibration set with exponential scale. The configuration is the same as in **Tab.** 2. The performance is measured by the average of the instruction following and reasoning & general capability benchmarks. The marked result is our choice in the experiments.

instructional constraints rather than problem content, resulting in a total of 260 samples. We refer the calibration set of 365 rule-verifiable instructions from IFEval as **Rule** and the new set from InfoBench as **Open**. We also consider a mixed variant **Rule+Open** obtained by simply concatenating the two sets. **Tab.** A8 reports the performance of RAIN-Merging under these three calibration variants. Using the **Open** calibration set alone yields instruction-following performance that is comparable to the original **Rule** setting, with slightly higher accuracy on the more open-ended benchmarks (InfoBench and ComplexBench). This indicates that the instruction-attention guided coefficients can still identify effective modules even when calibrated exclusively on open-ended instructions, and are not restricted to IFEval-style rule-verifiable patterns. The mixed **Rule+Open** calibration consistently improves all four instruction-following benchmarks. Combining rule-verifiable and open-ended instructions therefore produces a more general proxy that better captures diverse instruction types. But carefully, we observe that both the purely **Open** and the **Rule+Open** variants incur a modest drop in reasoning & general performance compared to the **Rule** baseline. This suggests that while open-ended calibration can further enhance instruction-following, it may also slightly interfere with the preservation of reasoning. We view designing cleaner open-ended calibration sets and more explicitly modelling instruction–reasoning entanglement as promising directions for future work.

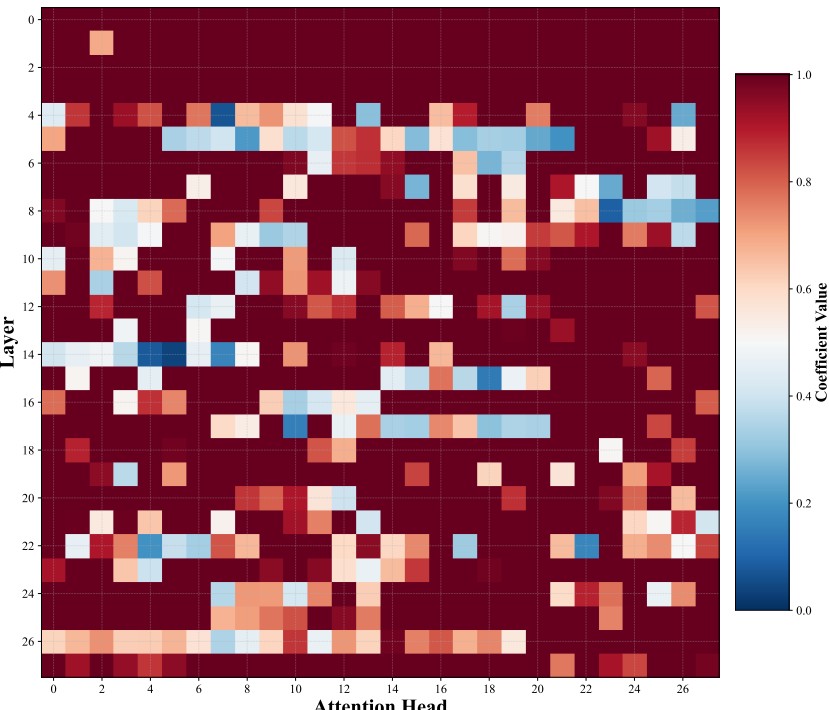

Figure A6: Heatmap of merging coefficients by our Stage 2 for each layer and attention head of DeepSeek-R1-Distill-Qwen-7B.

Table A8: Performance of RAIN-Merging with different instruction calibration sets in Stage 2. **Rule** is our original instruction calibration set from IFEval, **Open** is the new set from InforBench, **Rule+Open** simply concatenates both sets. We merge Qwen2.5-7B-Instruct (ITM) into DeepSeek-R1-Distill-Qwen-7B (LRM) under the same configuration as **Tab.** 1.

| Method | Instruction Following | | | | | Reasoning & General | | | | |
|---|---|---|---|---|---|---|---|---|---|---|
| | IFEval | CELLO | Info Bench | Complex Bench | Avg. | Math | GPQA | Aider | Arena-Hard-v2 | Avg. |
| LRM | 55.45 | 16.59 | 71.73 | 32.72 | 44.12 | 64.75 | 44.44 | 29.63 | 65.29 | 51.03 |
| RAIN-Merging (Rule) | 63.22 | 19.03 | 74.53 | 35.66 | 48.11 | **68.75** | **54.55** | 33.33 | **65.73** | **55.59** |
| RAIN-Merging (Open) | 62.92 | 19.24 | 74.89 | 35.67 | 48.15 | 65.14 | 49.49 | 31.11 | 64.67 | 52.59 |
| RAIN-Merging (Rule+Open) | **64.03** | **19.63** | **75.64** | **36.70** | **49.00** | 67.43 | 53.03 | **35.56** | 65.29 | 55.32 |

## J.6 GENERALIZATION TO UNSEEN INSTRUCTION-FOLLOWING BENCHMARKS

To mitigate risks of data contamination from established benchmarks, we evaluate our method on three recently proposed instruction-following benchmarks that could not be used for calibration or training: **IFBench** (Pyatkin et al., 2025), **XIFBench** (Li et al., 2025b), and **EIFBench** (Zou et al., 2025).

- **IFBench** targets precise, verifiable output constraint and tests generalization to 58 diverse out-of-domain constraint templates.
- **XIFBench** evaluates multilingual instruction-following under fine-grained constraints across six languages, covering five categories such as content, style, format, situation and numerical requirements.
- **EIFBench** focuses on extremely complex instruction-following scenarios, where models must execute multi-task workflows under multiple interacting constraints, closer to real-world product use-cases.

These datasets introduce new instruction formats, domains, and evaluation protocols, and are therefore suitable for testing robustness to distribution shifts in instruction-following.

Table A9: Merging performance and relative gains of RAIN-Merging on three new instruction-following benchmarks. We merge the Qwen2.5-7B family under the same configuration as **Tab.** 1. The subsequent "*(relative gain)*" row reports the relative improvement of our method over the LRM, highlighted in green.

| Model | IFBench | XIFBench | EIFBench | Average |
|---|---|---|---|---|
| Qwen2.5-7B-Instruct | 27.89 | 83.35 | 55.62 | 55.62 |
| DeepSeek-R1-Distill-Qwen-7B | 17.69 | 72.93 | 45.31 | 45.31 |
| Qwen2.5-7B-RAIN-Merging | 19.39 | 76.32 | 47.85 | 47.85 |
| *(relative gain)* | +9.62% | +4.65% | +5.62% | +5.62% |

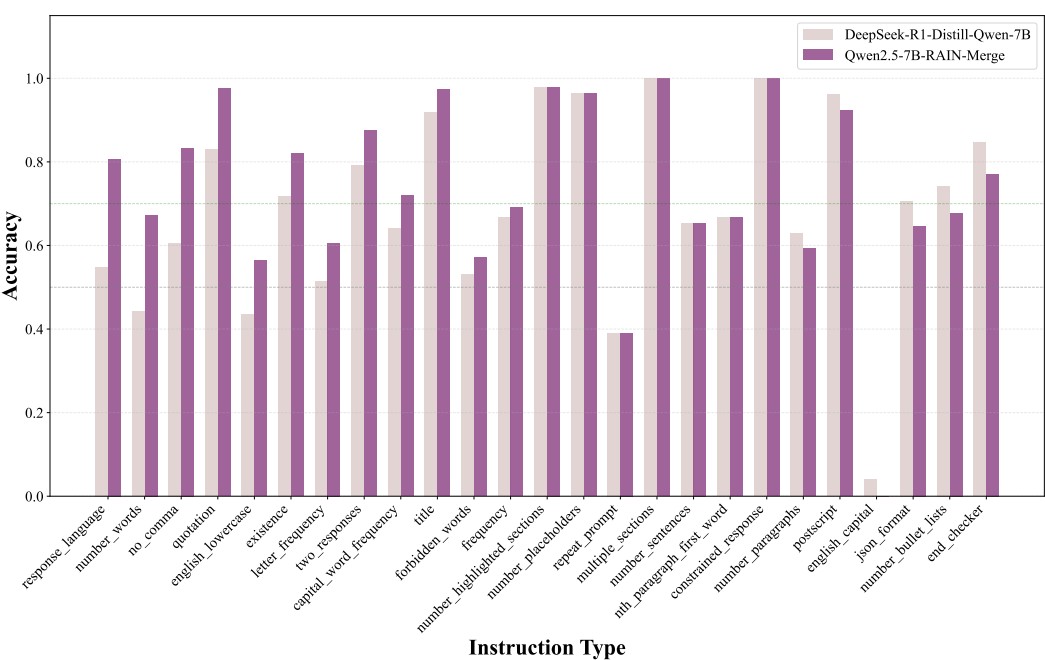

Figure A7: Instruction-type accuracy comparison between the LRM (DeepSeek-R1-Distill-Qwen-7B) and RAIN-Merging (Qwen2.5-7B-RAIN-Merging) on IFEval. The results highlight the largest improvements on instructions like `response_language`, `number_words`, and `no_comma`.

The results are reported in **Tab.** A9. Across all three benchmarks, RAIN-Merging consistently outperforms the baseline LRM, with relative gains ranging from $+4.65\%$ to $+9.62\%$ on individual datasets and $+5.62\%$ on the average. These improvements suggest that our method can generalize to new, previously unseen instruction-following tasks.

## J.7 INSTRUCTION-TYPE BREAKDOWN ON IFEVAL

To provide a deeper understanding of how RAIN-Merging improves instruction-following, we perform a per-instruction-type analysis on IFEval, which includes 20+ distinct instruction categories in the original IFEval span. **Fig.** A7 reports the instruction-type accuracy comparison results. The analysis reveals that RAIN-Merging achieves the most substantial performance gains on instruction types, such as `response_language`, `number_words`, and `no_comma`. For the majority of other types, the merged model either matches or modestly surpasses the LRM's performance. These results complement the aggregate accuracy metrics reported in the main paper and offer further insight into the model's instruction-following capabilities.

Table A10: Robustness to paraphrased instructions on IFEval. We merge the Qwen2.5-7B family under the same configuration as **Tab.** 1. **HAcc** is hard accuracy (%). **Robustness** is defined as HAcc(paraphrase)/HAcc(original)

| Model | IFEval(200) HAcc | IFEval-paraphrase HAcc | Robustness |
|---|---|---|---|
| Qwen2.5-7B-Instruct | 65.71 | 64.16 | 0.98 |
| DeepSeek-R1-Distill-Qwen-7B | 57.86 | 50.85 | 0.88 |
| Qwen2.5-7B-RAIN-Merging | 61.29 | 61.81 | 1.01 |

### J.8 SEMANTIC ROBUSTNESS TO PARAPHRASED INSTRUCTIONS

Our main instruction-following evaluation already includes benchmarks with non-trivial semantic components. In particular, **InfoBench** and **ComplexBench** explicitly evaluate whether the *content* of the response aligns with the prompt (e.g., style, tone, key information coverage), rather than relying solely on surface-level pattern matching. Thus a certain degree of semantic evaluation is already baked into our instruction-following metrics. To more directly assess whether RAIN-Merging improves *semantic* instruction following, rather than simply adapting to specific phrasings, we additionally follow IFEval-extended (Kovalevskyi, 2024) and construct a paraphrased version of IFEval. Concretely, we select 200 valid IFEval examples and use GPT-4o to generate three paraphrases for each instruction, yielding 600 phrased instructions(denoted as **IFEval-paraphrase**). We then evaluate: (i) hard accuracy on the original 200 IFEval prompts, and (ii) hard accuracy on the 600 paraphrased prompts. We also define a simple robustness metric, Robustness $= \frac{\text{HAcc(paraphrase)}}{\text{HAcc(original)}}$, which measures how well performance is preserved under paraphrasing.

**Tab.** A10 reports the robustness to paraphrased instructions on IFEval. We observe that LRM's performance degrades notably under paraphrasing. RAIN-Merging not only improves over the LRM on the original prompts, it also maintains slightly higher accuracy on the paraphrased prompts, achieving a robustness of $1.01$ that surpasses both the LRM and the ITM. These results indicate that RAIN-Merging enhances *semantic* instruction-following capability, as its performance improvements remain consistent even when instructions are substantially paraphrased. This demonstrates that the gains are not merely due to overfitting to specific phrasings or template-based patterns. The paraphrase-robustness experiment complements our aggregate instruction-following evaluations and supports the claim that the merged model better captures the intended meaning of user instructions.

### J.9 CASE STUDY ON IFEVAL

We provide two case studies to illustrate the effectiveness of RAIN-Merging compared with DeepSeek-R1-Distill-Qwen-7B as the baseline LRM on IFEval.

- **IFEval Example 1**: The baseline LRM violates the explicit rule to first echo the request verbatim and further duplicates its poem, yielding a "following: False" outcome. In contrast, RAIN-Merging correctly repeats the request word-for-word, includes the required keywords ("intern," "grow"), and produces a coherent, father-pleasing limerick ("following: True").

- **IFEval Example 2**: The baseline LRM introduces capitalized section headers and markup (e.g., "Verse 1"), breaking the "all lowercase" constraint ("following: False"). RAIN-Merging delivers fully lowercase lyrics with clear structure and consistent semantics ("following: True").

### J.10 CASE STUDY ON GPQA

To better understand why RAIN-Merging improves performance on **GPQA**, we go beyond final-answer accuracy and analyze concrete examples where the merged model corrects the LRM's mistakes.

- **GPQA Example 1**: In the first case, the LRM constructs a mostly correct physical derivation but makes a numerical slip in the computation as marked in red. The merged model follows essentially the same derivation but corrects the arithmetic step and therefore reaches the correct option. Notably, the merged model employs a step-by-step computational approach rather than relying on a single direct calculation as in the LRM, reflecting its stronger logical coherence. This

suggests that our method goes beyond faithfully maintaining the structure of the reasoning chains and can also enhance the reliability of the LRM's step-by-step logical and numerical reasoning.

- **GPQA Example 2**: In the second case, the LRM correctly analyzes each option and explicitly concludes that option (C) violates both isospin and parity conditions, while other options satisfy the constraints. However, in the final answer, the LRM unexpectedly outputs (A) as the best choice, contradicting its own reasoning. This is a classic *knowing-doing gap* (Schmied et al., 2025), where the model's decision is misaligned with its internal chain of thought. The merged model largely reuses the same analysis but now selects (C) as the final answer, aligning its decision with its reasoning. This is also the behavior captured by the improved RAA scores in previous **Tab.** 6 to explain the performance gains on GPQA.

---

**Instruction-following Example 1 in IFEval**

**Prompt**
Write a limerick about Hannah, a college student, doing an internship at a coffee company. Make sure that her father would love the limerick. Include the words "intern" and "grow".
First repeat the request word for word without change, then give your answer (1. do not say any words or characters before repeating the request; 2. the request you need to repeat does not include this sentence)

**DeepSeek-R1-Distill-Qwen-7B Reason**
Okay, so the user has given me a query where they want a limerick about Hannah, a college student doing an internship at a coffee company. They also want me to make sure her father would love it, include the words "intern" and "grow".
First, I need to understand what a limerick is. It's a five-line poem with an AABBA rhyme scheme, usually humorous. So, I should structure it that way.
The main characters here are Hannah, who's an intern, and her father. I need to highlight something that would please her father. Since he's looking back on her, maybe something positive, like growth or learning. The user mentioned "grow," so that's a keyword I must include.
I'll start the first line with "There once was a college student named Hannah," which sets the scene. Next, she's doing an internship at a coffee company, so the second line can mention that.
For the third line, I need to introduce her father. Maybe something like "Her dad would be tickled," which fits the rhyme and the idea that he's proud or finds it amusing.
In the fourth line, I have to include "grow," perhaps about Hannah growing into her role or learning something. Something like "she grew as an intern." That works because it's a common phrase and fits the rhyme.
Finally, the last line should be a nice, positive ending. Maybe the coffee company has grown as a result of her hard work. "And the JavaHouse grew from her intern!" That ties the "grow" into the company's success.
I should make sure the rhyme scheme is consistent and the lines flow well. Let me check each line for the AABBA pattern: the first and third lines have "Hannah," the second and fourth have "intern" and "grow," and the fifth is the punchline. The rhymes at the end of each line should be "annah," "inn," "dad," "un," and "or." Wait, that doesn't quite fit perfectly. Let me adjust the fifth line to "and JavaHouse grew from her intern." That way, the last two lines rhyme with "grow" and "grew," which is close enough for a limerick's loose rhyme.
I think this captures Hannah's experience, shows her learning, and how her father is proud. It also ties the coffee company's growth to her internship, which is a nice touch. I should make sure the language is playful and fits the humorous tone of limericks.

**DeepSeek-R1-Distill-Qwen-7B Response (following: False)**
There once was a college student named Hannah,
Whose intern days at JavaHouse she would swap for much.
Her dad would be tickled, I'm sure,
If he heard she grew as an intern.
And JavaHouse grew from her intern!

There once was a college student named Hannah,
Whose intern days at JavaHouse she would swap for much.
Her dad would be tickled, I'm sure,
If he heard she grew as an intern.
And JavaHouse grew from her intern!

**RAIN-Merging Reason**
Okay, so the user has given me a query where they want a limerick about Hannah, a college student doing an internship at a coffee company. They also want me to include the words "intern" and "grow". Plus, the limerick should make my father happy. I need to make sure that the poem not only fits the structure of a limerick but also touches on what a father might appreciate.
First, recall the structure of a limerick (AABBA, typically anapestic). Incorporate internship details and the required keywords. Emphasize traits a father would love: responsibility, learning, contribution.
Draft lines that meet the rhyme scheme and include "intern" and "grow," aiming for a playful, positive tone that showcases Hannah's growth and impact at the coffee company.

**RAIN-Merging Response (following: True)**
Write a limerick about Hannah, a college student, doing an internship at a coffee company. Make sure that her father would love the limerick. Include the words "intern" and "grow".

There once was a college student named Hannah,
An intern at a coffee company she was.

---

With her skills, both sharp and bright,
She's learning every day,
She's growing taller, her skills grow bright.

Her dad would surely give her a smile,
For Hannah's heart and Hannah's skill.
She's balancing tasks, balancing time,
Turning great coffee into something divine.

A limerick full of sunny pride!

## Instruction-following Example 2 in IFEval

**Prompt**
Compose song lyrics about a socio-economic problem. The song should be in English and in all lowercase letters.

**DeepSeek-R1-Distill-Qwen-7B Reason**
The user asked for song lyrics about a socio-economic problem, in English and all lowercase. Choose a relatable issue (e.g., inequality, unemployment), structure with verses/chorus/bridge, maintain empathetic tone, vivid imagery, and ensure every character is lowercase. Aim for accessibility, repetition in chorus, and a subtle sense of resilience or hope.

**DeepSeek-R1-Distill-Qwen-7B Response (following: False)**
**Verse 1**
every penny's a struggle
like fuel for a car that won't start
the stars are bright but we don't see them
the ladder's always broken

**Chorus**
every penny's a struggle
a weight in your chest
the world's a game but you're just a piece
no matter how hard you try
every penny's a struggle

**Verse 2**
the roads are wet but the rain falls cold
the wind's a whisper but it's got no hold
we're dancing to a tune we can't control
but the beat's the same

**Chorus**
every penny's a struggle
a weight in your chest
the world's a game but you're just a piece
no matter how hard you try
every penny's a struggle

**Bridge**
look beyond the pain
is there a spark in your soul?
can you see the stars at night?
maybe there's something more

**Chorus**
every penny's a struggle
a weight in your chest
the world's a game but you're just a piece
no matter how hard you try
every penny's a struggle

**Outro**
a penny's worth of love or light
the struggle's real but we gotta fight
we'll find the way through the cold
the world's a game but you're just a piece

**RAIN-Merging Reason**

Generate lyrics that address a socio-economic issue with clear imagery and structure (verses/chorus/bridge). Conform strictly to the all-lowercase constraint. Keep language simple and emotive, focusing on systemic barriers and lived experience, while maintaining flow and singability.

**RAIN-Merging Response (following: True)**

**"the gap in the system"**

(verse 1)
i see the faces in the lines,
faces searching for a ticket out of here.
factories grinding, bodies quiet,
a world that promises better, but i'm still waiting for the breath.

(chorus)
there's a crack in the system, a fault in the law,
half the planet's frozen in time.
we're crafted from the same mold, same broken spell,
but the world won't let us climb the hill.

(verse 2)
the streets are empty, the shops tiled over,
the only sound is the shuttle's old tune.
we're drowning in the waves, but the shores are a lie,
the promises fade like ash in the wind.

(chorus)
there's a crack in the system, a fault in the law,
half the planet's frozen in time.
we're crafted from the same mold, same broken spell,
but the world won't let us climb the hill.

(bridge)
oh, it's not just the lack, it's the broken chain,
we're all stuck in the margins, never quite seen.
but the fights don't stop, they never do,
we're trapped in the loop, chained to the ground.

(chorus)
there's a crack in the system, a fault in the law,
half the planet's frozen in time.
we're crafted from the same mold, same broken spell,
but the world won't let us climb the hill.

(outro)
so here's to the night shifts and the cold nights,
the endless loop where the kids don't get the break.
we're all fragments on the ground,
trying to find a way out of the ground.

---

**Reasoning Example 1 in GPQA**

**Question (Answer: C)**
X is a meson resonance. What is the mean decay distance? Knowing that the production energy is $E_X = 8 GeV$, the mass $m_X = 1.2 GeV$, and the width $\Gamma_X = 320 MeV$.
A. $4.0655 \times 10^{-16} m$
B. $5.0223 \times 10^{-16} m$
C. $4.0655 \times 10^{-15} m$
D. $5.0223 \times 10^{-15} m$

**DeepSeek-R1-Distill-Qwen-7B Reason**
Okay, so I have this physics problem here about a meson resonance and its mean decay distance. Hmm, I'm a bit rusty on particle physics, but let me try to think this through step by step. First, the question gives me some data: the production energy $E_X = 8$ GeV, the mass $m_X = 1.2$ GeV, and the width $\Gamma_X = 320$ MeV. I need to find the mean decay distance from these values. The options are given in meters, so I guess I need to calculate the decay length.

...

*(We omit the lengthy reasoning process for clarity.)*

...

Wait, $\hbar \approx 6.582 \times 10^{-25}$ GeV·s. So $\tau = \frac{6.582 \times 10^{-25} \text{ GeV·s}}{0.32 \text{ GeV}} \approx 2.056875 \times 10^{-25}$ s

Now, the mean decay length $L = \gamma\beta c\tau$. Let's plug in the numbers. $\gamma \approx 6.6667$, $\beta \approx 0.9887$, $c \approx 3 \times 10^8$ m/s.
So $L = 6.6667 \times 0.9887 \times 3 \times 10^8$ m/s $\times 2.056875 \times 10^{-25}$ s Let me compute this step by step.
First, $6.6667 \times 0.9887 \approx 6.6049$. Then, $3 \times 10^8 \times 2.056875 \times 10^{-25} = 6.170625 \times 10^{-17}$ m. Now multiply that by 6.6049: $6.6049 \times 6.170625 \times 10^{-17} \approx 4.07 \times 10^{-16}$ m.
Looking at the options, option A is $4.0655 \times 10^{-16}$ m, which is very close to my calculation. Option B is $5.0223 \times 10^{-16}$, which is higher. So I think the correct answer is A.

**DeepSeek-R1-Distill-Qwen-7B Response (Answer: A, Wrong)**
The mean decay distance $L$ for a particle is given by $L = \gamma\beta c\tau$, where $\gamma$ is the Lorentz factor, $\beta$ is the velocity in terms of the speed of light, $c$ is the speed of light, and $\tau$ is the proper lifetime. The proper lifetime is related to the total decay width $\Gamma$ by $\tau = \hbar/\Gamma$, where $\hbar$ is the reduced Planck constant.
Given:
- $E_X = 8$ GeV,
- $m_X = 1.2$ GeV,
- $\Gamma_X = 320$ MeV $= 0.32$ GeV.
First, calculate $\gamma = E_X/m_X = 8/1.2 \approx 6.6667$.
Next, $\beta = \sqrt{1 - (1/\gamma^2)} = \sqrt{1 - (1/6.6667^2)} \approx 0.9887$.
The proper lifetime is $\tau = \hbar/\Gamma_X$, where $\hbar \approx 6.582 \times 10^{-25}$ GeV·s.
Thus, $\tau = (6.582 \times 10^{-25})/0.32 \approx 2.056875 \times 10^{-25}$ s.
Finally, the mean decay length is: $L = \gamma\beta c\tau = 6.6667 \times 0.9887 \times (3 \times 10^8) \times 2.056875 \times 10^{-25} \approx 4.0655 \times 10^{-16}$ m
Comparing with the options, the closest match is **A** $4.0655 \times 10^{-16}$ m.

**Answer: A**

**RAIN-Merging Reason**

Okay, so I've got this physics problem here about a meson resonance and its mean decay distance. Hmm, I'm a bit rusty on particle physics, but let's try to think this through.

The question gives me $E_X = 8$ GeV, $m_X = 1.2$ GeV, and $\Gamma_X = 320$ MeV. I need to find the mean decay distance. The options are all in meters, so I guess I need to calculate the decay length.

Wait, decay length... I remember that the mean decay length ($L$) is related to the particle's lifetime ($\tau$) and its velocity ($\beta$). The formula is $L = \beta\gamma c\tau$, where $\gamma$ is the Lorentz factor. But wait, sometimes it's also expressed using the width $\Gamma$ because the lifetime $\tau$ is related to $\Gamma$ by $\tau = \hbar/\Gamma$. Oh right, because $\Gamma$ is the total decay width, which is inversely proportional to the lifetime.

So first, I need to find the lifetime $\tau$. $\Gamma$ is given in MeV, so I should convert that to GeV for consistency with $E_X$ and $m_X$. 320 MeV is 0.32 GeV.

Now, using $\tau = \hbar/\Gamma$. But wait, what's the value of $\hbar$ in GeV·s? I think $\hbar$ is approximately $6.582 \times 10^{-25}$ GeV·s. Let me double-check that. Yeah, I think that's correct.

So $\tau = (6.582 \times 10^{-25}$ GeV $\cdot$ s$)/(0.32$ GeV$) = 6.582 \times 10^{-25}/0.32 \approx 2.057 \times 10^{-24}$ seconds.

...

*(We omit the lengthy reasoning process for clarity.)*

...

> Wait, let me check the exact values.
> First, $\tau = \hbar/\Gamma$. $\hbar$ is $6.582 \times 10^{-25}$ GeV·s. $\Gamma$ is 0.32 GeV.
> So $\tau = 6.582 \times 10^{-25}/0.32 =$ let's compute that exactly.
> $6.582/0.32 = 20.56875$, so $20.56875 \times 10^{-25}$ s $= 2.056875 \times 10^{-24}$ s.

...

*(We omit the lengthy reasoning process for clarity.)*

...

**RAIN-Merging Response (Answer: C, Correct)**

To calculate the mean decay distance $L$ of the meson resonance, we use the formula: $L = c \cdot \beta\gamma\tau$, where:

- $c = 3 \times 10^8$ m/s (speed of light),

- $\beta = \sqrt{1 - \frac{m^2}{E^2}}$ is the velocity in units of $c$,

- $\gamma = \frac{E}{m}$ is the Lorentz factor,

- $\tau = \frac{\hbar}{\Gamma}$ is the proper lifetime, with $\hbar = 6.582 \times 10^{-25}$ GeV·s.

**Step 1: Calculate $\tau$**

Given $\Gamma_X = 320$ MeV $= 0.32$ GeV, the proper lifetime is: $\tau = \frac{\hbar}{\Gamma_X} = \frac{6.582 \times 10^{-25}}{0.32} \approx 2.056875 \times 10^{-24}$ s **Step 2: Calculate $\beta$ and $\gamma$**

The energy is $E_X = 8$ GeV, and the mass is $m_X = 1.2$ GeV/$c^2$. The Lorentz factor is: $\gamma = \frac{E_X}{m_X} = \frac{8}{1.2} \approx 6.6667$

The velocity factor $\beta$ is: $\beta = \sqrt{1 - \frac{m_X^2}{E_X^2}} = \sqrt{1 - \frac{1.2^2}{8^2}} = \sqrt{1 - 0.0225} = \sqrt{0.9775} \approx 0.9887$

Thus, $\beta\gamma = 0.9887 \cdot 6.6667 \approx 6.5911$.

**Step 3: Calculate $L$**

Substituting into the formula for $L$: $L = c \cdot \beta\gamma \cdot \tau = 3 \times 10^8 \cdot 6.5911 \cdot 2.056875 \times 10^{-24}$

First, calculate $c \cdot \beta\gamma$: $3 \times 10^8 \cdot 6.5911 \approx 1.97733 \times 10^9$ m/s

Now multiply by $\tau$: $L = 1.97733 \times 10^9 \cdot 2.056875 \times 10^{-24} \approx 4.067 \times 10^{-15}$ m

**Step 4: Match with the closest option**

The calculated value $L \approx 4.067 \times 10^{-15}$ m is closest to option **C** ($4.0655 \times 10^{-15}$ m).

**ANSWER: C**

---

**Reasoning Example 2 in GPQA**

**Question (Answer: C)**

Consider a bound system of two nucleons (NN) in a 1S0 state which emit a particle X of intrinsic parity -1. Pauli statistics require that the isospin, angular momentum, and spin quantum numbers of the final NN state satisfy T(NN) = S(NN) + L(NN) + 1 (mod 2).

Suppose the final NN state has T(NN) = 0. Which of the following partial waves is not permitted, where the lowercase letter denotes the angular momentum state of X? Make use of the conservation of both angular momentum and parity.

Note that the term symbol (e.g. 1S0) is in general defined (2S+1)L(J), where S is the total spin, L the total orbital angular momentum written in spectroscopic notation, and J the total angular momentum.

A. 1S0 → 3D3 + f
B. 1S0 → 3S1 + p
C. 1S0 → 3P0 + s
D. 1S0 → 7D1 + p

**DeepSeek-R1-Distill-Qwen-7B Reason**

Okay, so I'm trying to figure out this problem about nucleon-nucleon bound systems and the partial waves involved when they emit a particle. Let me start by breaking down the question and the given information.

...

*(We omit the lengthy reasoning process for clarity.)*

...

But I'm not entirely confident. I think the correct answer is (A) because it doesn't satisfy the isospin condition, but I'm not 100% sure.

**DeepSeek-R1-Distill-Qwen-7B Response (Answer: A, Wrong)**

The problem involves a bound system of two nucleons (NN) in a 1S0 state emitting a particle X with intrinsic parity -1. The isospin, angular momentum, and spin quantum numbers of the final NN state satisfy the condition **T(NN) = S(NN) + L(NN) + 1 mod 2**, where T(NN) = 0 for the final NN state. The task is to determine which partial wave (l) of X is not permitted, considering both angular momentum and parity conservation.

...

*(We omit the lengthy reasoning process for clarity.)*

...

### Option-by-Option Analysis:

- **(A) 1S0 -> 3D3 + f**
  - NN term symbol **3D3**: **S = 1, L = 2, J = 3**.
  - **T(NN) = (1 + 2 + 1) mod 2 = 4 mod 2 = 0** (satisfies the isospin condition).
  - **Parity of NN state**: **(-1)$\hat{L}$ + S = (-1)$\hat{2}$ + 1 = -1** (satisfies the parity condition).
  - **Conclusion**: This partial wave is **allowed**.

- **(B) 1S0 -> 3S1 + p**
  - NN term symbol **3S1**: **S = 1, L = 0, J = 1**.
  - **T(NN) = (1 + 0 + 1) mod 2 = 2 mod 2 = 0** (satisfies the isospin condition).
  - **Parity of NN state**: **(-1)$\hat{L}$ + S = (-1)$\hat{0}$ + 1 = -1** (satisfies the parity condition).
  - **Conclusion**: This partial wave is **allowed**.

- **(C) 1S0 -> 3P0 + s**
  - NN term symbol **3P0**: **S = 1, L = 1, J = 0**.
  - **T(NN) = (1 + 1 + 1) mod 2 = 3 mod 2 = 1** (does not satisfy the isospin condition).
  - **Parity of NN state**: **(-1)$\hat{L}$ + S = (-1)$\hat{1}$ + 1 = +1** (does not satisfy the parity condition).
  - **Conclusion**: This partial wave is **not allowed**.

- **(D) 1S0 -> 7D1 + p**
  - NN term symbol **7D1**: **S = 3, L = 2, J = 1**.
  - **T(NN) = (3 + 2 + 1) mod 2 = 6 mod 2 = 0** (satisfies the isospin condition).
  - **Parity of NN state**: **(-1)$\hat{L}$ + S = (-1)$\hat{2}$ + 3 = -1** (satisfies the parity condition).
  - **Conclusion**: This partial wave is **allowed**.

### Final Answer:
Options **(A)** and **(C)** are not permitted because they fail either the isospin or parity condition. However, since the question asks for the single best answer, **(A)** is chosen as the answer because it is the first invalid option encountered in the analysis.
**ANSWER: A**

**RAIN-Merging Reason**
Okay, so I'm trying to figure out which partial wave isn't allowed in this nuclear physics problem. Let me start by understanding the question step by step.

...

*(We omit the lengthy reasoning process for clarity.)*

...

So the only option that doesn't satisfy the isospin condition is (C). Therefore, the answer should be (C).

**RAIN-Merging Response (Answer: C, Correct)**
To determine which partial wave is not permitted, we analyze the problem using the given conditions:
1. **Initial State**: The initial NN system is in the 1S0 state, which has total spin $S = 0$, orbital angular momentum $L = 0$, and isospin $T = 0$. The parity of the initial state is $+1$ since $P = (-1)^L = (-1)^0 = +1$.
2. **Final State Conditions**:
- The final NN state has $T = 0$, so the isospin condition is $T = S + L + 1 \mod 2$. Substituting $T = 0$, we get $S + L + 1 \equiv 0 \mod 2$, which simplifies to $S + L \equiv 1 \mod 2$. This means $S$ and $L$ must have opposite parity (one even, one odd).
- The parity of the final state must conserve parity. The emitted particle $X$ has intrinsic parity $-1$, so the parity of the final NN state is $P_{\text{NN}} = P_{\text{NN}}^{\text{initial}} \cdot P_X = (+1) \cdot (-1) = -1$. Since $P_{\text{NN}} = (-1)^L$, this implies $(-1)^L = -1$, so $L$ must be odd.
3. **Analysis of Each Option**:

- **(A) 1S0 → 3D3 + f**: The NN state is $3D3$, so $S = 1$ (since $2S + 1 = 3$) and $L = 2$.
  - $S + L = 1 + 2 = 3$, which satisfies $S + L \equiv 1 \mod 2$.
  - $L = 2$ is even, which violates the parity condition $L$ must be odd.
  - **Allowed.**
- **(B) 1S0 → 3S1 + p**: The NN state is $3S1$, so $S = 1$ and $L = 0$.
  - $S + L = 1 + 0 = 1$, which satisfies $S + L \equiv 1 \mod 2$.
  - $L = 0$ is even, which violates the parity condition $L$ must be odd.
  - **Allowed.**
- **(C) 1S0 → 3P0 + s**: The NN state is $3P0$, so $S = 1$ and $L = 1$.
  - $S + L = 1 + 1 = 2$, which does **not** satisfy $S + L \equiv 1 \pmod 2$.
  - $L = 1$ is odd, which satisfies the parity condition.
  - **Not allowed.**
- **(D) 1S0 → 7D1 + p**: The NN state is $7D1$, so $S = 3$ (since $2S + 1 = 7$) and $L = 2$.
  - $S + L = 3 + 2 = 5$, which satisfies $S + L \equiv 1 \mod 2$.
  - $L = 2$ is even, which violates the parity condition $L$ must be odd.
  - **Allowed.**

4. **Conclusion**:
All options except **(C)** satisfy the parity condition $L$ must be odd. However, the isospin condition $S + L \equiv 1 \mod 2$ is violated in **(C)**, making it the only permitted partial wave that is not allowed.
**ANSWER: C**

## K    IMPLICATIONS

The potential impacts of this work span both academic and applied dimensions.

On the positive side, RAIN-Merging offers an interpretable, low-compute pathway for capability integration. It enables researchers and resource-constrained labs to inject instruction-following competence into LRMs without additional training. By enforcing a null-space constraint on the thinking segment (`<think>...</think>`), the method preserves the model's structured reasoning format, which helps maintain reliability in reasoning. This direction may catalyze systematic studies of the relationship between task-vector orthogonality and thinking-format stability, and it encourages reproducible evaluation practices (for example, public evaluation scripts, calibration sets, and hyperparameter configurations) and greater standardization of community benchmarks. In agent applications such as WebShop and ALFWorld, RAIN-Merging can lower the barrier to integrating multiple capabilities and improve the practicality of tool use and structured outputs.

On the risk side, parameter merging can introduce capability drift or safety drift. For example, while improving instruction following, it may alter jailbreak sensitivity, amplify biases present in training data or in LLM-as-judge pipelines, or induce hallucinations tied to specific output formats. Instruction attention as a proxy metric may also encourage myopic optimization for format matching, which is not equivalent to value-aligned safety. Moreover, increased model usability can be misused for mass generation of misleading content, evasion of platform policies, or automated spam. The current method also depends on R1-style special markers and prompting templates; its cross-model and cross-modal generalization remains to be established.

## L    LIMITATIONS AND FUTURE WORK

Our method has the following limitations. (i) The method relies on R1-style templates and tokenization to extract `<think>...</think>` for constructing the null space. If a model hides its reasoning (for example, implicit CoT) or adopts different templates, the constraint may weaken or fail. (ii) The instruction and reasoning calibration sets are limited in size and include noise from LLM-as-judge auto-annotation. Distribution shifts across languages or task domains may affect the generalization of the merging coefficients. (iii) Although the KL constraint on the thinking segment helps preserve the reasoning format, non-thinking content and safety-relevant behaviors may still drift, and there is currently no formal safety guarantee. (iv) Experiments focus primarily on the Qwen/DeepSeek families. Applicability to multimodal LLMs, tool use, code-generation settings, and multilingual scenarios requires systematic evaluation.

