# OpenReview forum: "RAIN-Merging: A Gradient-Free Method to Enhance Instruction Following in Large Reasoning Models with Preserved Thinking Format"
_ICLR.cc/2026/Conference — ICLR 2026 Oral_

### Official Review · Reviewer_2pjo · 2025-10-27

**Soundness:** 3
**Presentation:** 3
**Contribution:** 3
**Rating:** 8
**Confidence:** 3

**Summary:**

This paper uses task vectors to alleviate the issue of reasoning models where they are struggling with instruction following. Authors proposed a gradient-free method to address this problem, called Reasoning-Aware Instruction-attention guided Null-space projection Merging (RAIN-Merging). At the first stage, ITM task vector is projected on the null space of forward features at thinking tokens, and at the next stage, an instruction attention is estimated to further amplify instruction-relevant components.

**Strengths:**

1. The paper is well-written, and figures and tables help better understand the complicated method that authors proposed.
2. Reasoning models are bad at instruction-following is a well-known phenomenon, and the proposed method uses simple task vectors to boost the instruction-following ability while maintaining the original reasoning performance.
3. The results are reported in multi-dimensions according to the research questions in Section 4. Specifically, reporting a performance in agentic scenarios is a very good experimental evidence that the proposed method works well in the real world.
4. The proposed method is theoretically well-grounded.

**Weaknesses:**

1. In Table 1, there are some cases where RAIN-Merging outperforms the performance of the original LRM. Authors hypothesize that stronger instruction adherence improves CoT quality. -- I suggest to prove this hypothesis (by manually checking randomly sampled predictions). This phenomenon seems very interesting to me since my intuition says the opposite. Specifically, the performance is increased by ~10% in GPQA. How is it possible?
2. In the ablation study, only stage 2 is ablated. Therefore, the paragraph's name should be scoped down. Also, could you ablate stage 1 as well to prove its effectiveness?

Despite some of these questions, I believe this paper would contribute to the community significantly.

**Questions:**

See Weaknesses.

---

> ### Author Response · Authors · 2025-11-21
> **Response to Reviewer 2pjo (Part 1)**
>
> Thank you for your constructive feedback and for acknowledging the clarity of our presentation and the effectiveness of our method. Below we address your concerns in turn:
>
> **W2: Completing the Ablation on Both Stage-1 and Stage-2**
>
> > **Table R8**: Performance of ablation on stage-1 and stage-2. We merge Qwen2.5-7B-Instruct (ITM) into DeepSeek-R1-Distill-Qwen-7B (LRM) under the same setup as Tab.1 in manuscript.
>
> | Method                  | Instruction-following Avg. | Reasoning & General Avg. |
> |-------------------------|--------|--------|
> | LRM     | 44.12 | 51.03  |
> | RAIN-Merging w/o Stage 2| 46.58  | 54.92  |
> | RAIN-Merging w/o Stage 1| 47.62  | 52.44  |
> | RAIN-Merging     | 48.11  | 55.59  |
>
> > Thank you for pointing this out. We agree that a complete ablation should examine the contribution of both stages. In the revised version, we add the ablation where we disable stage-1 while keeping stage-2 active (RAIN-Merging w/o Stage 1). The results are shown in Tab.R7 and Tab.4 in the revised manuscript. We observe that w/o stage-1 improves instruction-following more than w/o stage-2 but comes at a noticeable cost to reasoning and general ability, consistent with the fact that stage-1 explicitly protects the thinking format. The full RAIN-Merging combines the strengths of both stages, achieving the best trade-off with the highest instruction-following and the highest reasoning and general performance. These results confirm that both stage-1 and stage-2 in our RAIN-Merging are necessary and complementary. We have updated the ablation section and included the above table results in the revised paper.
>
> **W1: Reasoning Quality Evaluation and Case Studies for Improvement on GPQA**
>
> > **Table R7**: CoT quality evaluation of reasoning traces and responses. We merge the Qwen2.5-7B family under the same configuration as Tab.1 in manuscript. We report Reasoning Internal Coherence (**RIC**) and Reasoning-Answer Alignment (**RAA**) on GPQA (0-5 scale). The subsequent “(*relative gain*)” row reports the relative improvement of our method over the LRM.
>
> | Model                       | RIC    | RAA     |
> | --------------------------- | ------ | ------- |
> | DeepSeek-R1-Distill-Qwen-7B | 3.53   | 3.76    |
> | Qwen2.5-7B-RAIN-Merging     | 3.56   | 4.26    |
> | (*relative gain*)           | +0.86% | +13.31% |
>
>
> > We thank the reviewer for noticing this phenomenon and for the suggestion to inspect the underlying CoT behavior. We also regard the improvement on GPQA intriguing and perform two analyses to better understand it.
> >
> > **(a) Reasoning and answer traces evaluation.**
> >
> > Following [1] we adopt two metrics to assess the quality of reasoning and answer traces (as in response to **Reviewer i2ye W3.1&W4.1&Q2**):
> > * Reasoning Internal Coherence (**RIC**): how logically consistent and self-contained the reasoning trace is.
> > * Reasoning–Answer Alignment (**RAA**): how well the reasoning trace semantically supports the final answer.
> >
> > We observe that RAIN-Merging almost preserves RIC, while substantially improving RAA ($+13.31\%$). This supports our hypothesis that stronger instruction-following makes the model more likely to produce responses that faithfully adhere to  reasoning traces.

---

> ### Author Response · Authors · 2025-11-21
> **Response to Reviewer 2pjo (Part 2)**
>
> > **(b) Manual case studies on GPQA.**
> >
> > We further manually inspect GPQA examples where RAIN-Merging corrects the LRM’s mistakes. Two representative cases are added to the Appendix:
> >
> > 1). In the first case, the LRM constructs a mostly correct physical derivation but makes a numerical slip in the computation, e.g.,
> > ```
> > Wait, ħ is approximately 6.582 x 10^-25 GeV·s. So τ = (6.582e-25 GeV·s) / (0.32 GeV) ≈ 2.056875e-25 seconds.
> > ```
> > which propagates to an incorrect final answer. The merged model follows the same derivation but corrects the arithmetic step,
> > ```
> > First, τ = ħ / Γ. ħ is 6.582e-25 GeV·s. Γ is 0.32 GeV.
> >
> > So τ = 6.582e-25 / 0.32 = let's compute that exactly.
> >
> > 6.582 / 0.32 = 20.56875, so 20.56875e-25 s = 2.056875e-24 s.
> > ```
> > and therefore reaches the correct option. This suggests that our method goes beyond faithfully maintaining the structure of the reasoning chains to also enhance the LRM's inherent logical capabilities.
> >
> > 2). In the second case, the LRM correctly analyzes each option and explicitly concludes that option (C) violates both isospin and parity conditions, yet unexpectedly outputs (A) as the final answer.
> > ```
> > ### Option-by-Option Analysis:
> > - **(A) 1S0 -> 3D3 + f**
> >   - NN term symbol **3D3**: **S = 1**, **L = 2**, **J = 3**.
> >   - **T(NN) = (1 + 2 + 1) mod 2 = 4 mod 2 = 0** (satisfies the isospin condition).
> >   - **Parity of NN state**: **(-1)^{L + S} = (-1)^{2 + 1} = -1** (satisfies the parity condition).
> >   - **Conclusion**: This partial wave is **allowed**.
> >
> > - **(B) 1S0 -> 3S1 + p**
> >   - NN term symbol **3S1**: **S = 1**, **L = 0**, **J = 1**.
> >   - **T(NN) = (1 + 0 + 1) mod 2 = 2 mod 2 = 0** (satisfies the isospin condition).
> >   - **Parity of NN state**: **(-1)^{L + S} = (-1)^{0 + 1} = -1** (satisfies the parity condition).
> >   - **Conclusion**: This partial wave is **allowed**.
> >
> > - **(C) 1S0 -> 3P0 + s**
> >   - NN term symbol **3P0**: **S = 1**, **L = 1**, **J = 0**.
> >   - **T(NN) = (1 + 1 + 1) mod 2 = 3 mod 2 = 1** (does not satisfy the isospin condition).
> >   - **Parity of NN state**: **(-1)^{L + S} = (-1)^{1 + 1} = +1** (does not satisfy the parity condition).
> >   - **Conclusion**: This partial wave is **not allowed**.
> >
> > - **(D) 1S0 -> 7D1 + p**
> >   - NN term symbol **7D1**: **S = 3**, **L = 2**, **J = 1**.
> >   - **T(NN) = (3 + 2 + 1) mod 2 = 6 mod 2 = 0** (satisfies the isospin condition).
> >   - **Parity of NN state**: **(-1)^{L + S} = (-1)^{2 + 3} = -1** (satisfies the parity condition).
> >   - **Conclusion**: This partial wave is **allowed**.
> >
> > ### Final Answer:
> > Options **(A)** and **(C)** are not permitted because they fail either the isospin or parity condition. However, since the question asks for the single best answer, **(A)** is chosen as the answer because it is the first invalid option encountered in the analysis.
> >
> > **ANSWER: A**
> > ```
> >
> > This is a classic “knowing–doing gap” [2], where the model’s decision contradicts its own reasoning. The merged model reuses almost the same analysis but now selects (C) as the final answer, aligning its decision with its reasoning. This is exactly the behavior captured by the improved RAA scores: RAIN-Merging does not fundamentally change the underlying reasoning chain, but makes the model adhere more faithfully to its reasoning traces.
> >
> > Taken together, these CoT metrics and case studies indicate that the GPQA gains are not accidental. RAIN-Merging preserves the LRM’s reasoning structure, while enhanced instruction-following reduces errors such as inconsistent final choices or sloppy arithmetic. In other words, instruction-following here acts less as “stronger reasoning” and more as a mechanism that enforces consistency between the internal chain of thought and the external answer. We have added the GPQA RAA/RIC scores in Appendix J.9 and the above case studies (Pages 37–40) in Appendix J.13 of the revised paper, and we highlight this “instruction–reasoning mutual enhancement” effect as an interesting direction for future work.
>
> ### Reference:
>
> [1] Wang, Changyue, et al. "Joint Evaluation of Answer and Reasoning Consistency for Hallucination Detection in Large Reasoning Models." arXiv 2025.
>
> [2] Schmied, Thomas, et al. "Llms are greedy agents: Effects of rl fine-tuning on decision-making abilities." arXiv 2025.

---

> > ### Comment · Reviewer_2pjo · 2025-11-24
> >
> > Thanks for the responses. Most of my concerns are addressed. BTW, the case study on GPQA looks very interesting. I think these results align well to the research question of the paper. I'll keep my score the same. Great work!

---

> > > ### Author Response · Authors · 2025-11-25
> > >
> > > Thank you very much for your kind follow-up. We’re happy that our response resolved most of your concerns and that the GPQA case study resonated with you. We truly appreciate your encouraging comments and positive evaluation.

---

### Official Review · Reviewer_i2ye · 2025-10-31

**Soundness:** 3
**Presentation:** 3
**Contribution:** 3
**Rating:** 4
**Confidence:** 4

**Summary:**

This paper introduces RAIN-Merging, a gradient-free method for enhancing instruction following in Large Reasoning Models while preserving their structured thinking format. The approach projects instruction task vectors onto the null space of thinking token features, then applies instruction-attention guided scaling coefficients. The method demonstrates improvements across instruction-following and reasoning benchmarks without requiring gradient-based training.

**Strengths:**

1. Merging ITM and LRM is an interesting and practical problem.
2. The finding that task vectors' principal subspaces are nearly orthogonal across key modules provides an interesting understanding about the parameter space structure of these capabilities.
3. The evaluation spans multiple model families and sizes.
4. The gradient-free nature makes this a practical, accessible alternative to SFT.
5. Using four instruction-following benchmarks and multiple reasoning datasets provides reasonable empirical breadth.

**Weaknesses:**

### 1. Data Contamination / Generalization Concerns
For example, Qwen2.5-7B-Instruct is trained on IFEval, InfoBench, and ComplexBench as calibration data, and this paper evaluates RAIN-Merging on the same benchmarks. Results may not generalize to unseen instruction-following or reasoning scenarios. Maybe the null-space projection and coefficients are optimized on the same distribution they're tested on.

### 2. Data
The paper evaluates instruction-following and reasoning on separate benchmark datasets. While the paper attempts to evaluate integrated capabilities using agentic scenarios in Table 3, these tasks may not simultaneously stress complex reasoning and strict, arbitrary instruction-following to the same degree as the benchmarks. The main evaluation in Table 1 still separates these two skills, leaving a gap in the core claim.

### 3. Metric
The paper relies primarily on accuracy metrics across all benchmarks. More metrics would be valuable to answer such questions: Is the thinking process coherent, or just structurally preserved (only thinking tokens)? Which types of instructions improve most/least? Are outputs semantically following instructions, or just superficially (evaluation on phrased instructions)?

### 4. Method
In stage 1, how do we know null-space projection preserves reasoning ability rather than just token usage? Does the model still perform meaningful reasoning in <think> blocks, or just maintain the format while reasoning quality degrades? If the reasoning benchmark results show preservation, but is this because the reasoning content is truly preserved, or the model learned to use thinking tokens without meaningful reasoning?

In stage 2, what happens when reasoning and instruction-following are highly entangled? How are attention-guided coefficients computed when both reasoning and instruction-following are active in the same tokens?

**Questions:**

* What is your hypothesis for why reasoning ability and instruction-following have low coupling in parameter space? Is this due to structural differences (thinking tokens vs. output format) rather than semantic content differences?
* How can you verify that Stage 1 preserves actual reasoning ability (content) and not just the habit of using thinking tokens? What metrics or analyses distinguish these scenarios?
* In Stage 2, how do attention-guided coefficients handle tokens where reasoning and instruction-following are deeply entangled (e.g., instructions about reasoning strategy, logical constraints, or structured argumentation)? Can you provide examples and analysis?

---

> ### Author Response · Authors · 2025-11-21
> **Response to Reviewer i2ye (Part 1)**
>
> Thank you for your constructive feedback and for recognizing our practical value and broad evaluation. Below we address your concerns in turn:
>
> **W1: Data Contamination / Generalization Concerns**
>
> > **Table R3**: Merging performance and relative gains of RAIN-Merging on three new instruction-following benchmarks. We merge the Qwen2.5-7B family under the same configuration as Tab.1 in manuscript. The subsequent “(*relative gain*)” row reports the relative improvement of our method over the LRM.
>
> | Model                          | IFBench | XIFBench | EIFBench | Average |
> |---------------------------------|---------|----------|----------|---------|
> | Qwen2.5-7B-Instruct             | 27.89   | 83.35    | 55.62    | 55.62   |
> | DeepSeek-R1-Distill-Qwen-7B    | 17.69   | 72.93    | 45.31    | 45.31   |
> | Qwen2.5-7B-RAIN-Merging        | 19.39   | 76.32    | 47.85    | 47.85   |
> | (*relative gain*)              | +9.62% | +4.65% | +5.62% | +5.62% |
>
> > Thank you for raising this important concern about data contamination and generalization. To address this, we have conducted additional experiments on three recently proposed instruction-following benchmarks to evaluate our method's performance on unseen data and avoid potential data contamination. These datasets are IFBench[1], XIFBench[2], and EIFBench[3].
> >
> > As shown in Tab.R3 and Tab.A10 (Appendix J.7), our method provides consistent improvements over the baseline LRM, with performance gains ranging from 4.65% to 9.62% across all three new benchmarks. This validates that RAIN-Merging can generalize to new, previously unseen instruction-following tasks. This new experiment and the corresponding discussion have been added to the revised Appendix J.7.
>
> **W2: Joint Evaluation of Reasoning and Instruction-Following**
>
> > **Table R4**: Merging performance and relative gains of RAIN-Merging on MathIF[4]. We merge the Qwen2.5-7B family under the same configuration as Tab.1 in manuscript. **IF Acc.** is the hard accuracy of satisfying all instruction constraints, **Math Acc.** is math accuracy under constraints, and **Both Acc.** is the fraction of samples where both constraints and math answers are correct. The subsequent “(*relative gain*)” row reports the relative improvement of our method over the LRM.
>
> | Model                         | IF Acc. | Math Acc. | Both Acc. |
> |-------------------------------|-------------|--------------|--------------|
> | Qwen2.5-7B-Instruct           | 48.81       | 40.95        | 19.76        |
> | DeepSeek-R1-Distill-Qwen-7B   | 25.86       | 53.81        | 12.62        |
> | Qwen2.5-7B-RAIN-Merging       | 35.10       | 54.76        | 20.48        |
> | (*relative gain*)    | +35.73%   | +1.77%     | +62.26%    |
>
> > We appreciate this insightful comment. We primarily evaluate instruction-following and reasoning on separate benchmarks. To more directly assess the joint capability of instruction-following and reasoning, we additionally evaluate RAIN-Merging on MathIF[4]. MathIF is explicitly designed to measure instruction-following in mathematical reasoning tasks. This benchmark is therefore well aligned with our goal of assessing whether a merged model can simultaneously maintain strong reasoning and obey instructions in a single task.
> >
> > Results are shown in Tab.R4 and Tab.A11 (Appendix J.8). Compared to the LRM, RAIN-Merging improves instruction-following while keeping math reasoning. Most importantly, on the joint metric **Both Acc**, which requires simultaneous success in reasoning and instruction following, RAIN-Merging increases performance from $12.62\% \rightarrow 20.48\%$ ($\approx+62.3\%$ relative), outperforming both the LRM and the ITM.
> >
> > These results indicate that RAIN-Merging not only improves instruction-following over the LRM, but also preserves its reasoning accuracy, leading to a substantial gain on the core target of "both correct and follow". We have added these experiments and the corresponding discussion to the revised Appendix J.8.

---

> ### Author Response · Authors · 2025-11-21
> **Response to Reviewer i2ye (Part 2)**
>
> **W3.1&W4.1&Q2: Reasoning and Answer Traces Evaluation**
>
> > **Table R5**: Evaluation of reasoning and answer traces. We merge the Qwen2.5-7B family under the same configuration as Tab.1 in manuscript. We report Reasoning Internal Coherence (**RIC**) and Reasoning-Answer Alignment (**RAA**) on IFEval, AIME25, and GPQA (0-5 scale). The subsequent “(*relative gain*)” row reports the relative improvement of our method over the LRM.
>
> | Model | IFEval RIC | IFEval RAA | AIME25 RIC | AIME25 RAA | GPQA RIC | GPQA RAA | Average |
> |------------------------------|------------|------------|------------|------------|----------|----------|------------|
> | DeepSeek-R1-Distill-Qwen-7B  | 4.58       |  4.41      | 4.50       |  3.60       | 3.53     |  3.76   | 4.06       |
> | Qwen2.5-7B-RAIN-Merging      | 4.61      |  4.51       | 4.50       |  4.10     | 3.56    | 4.26    | 4.26       |
> | (*relative gain*)   | +0.77%  | +2.26% | 0.00% | +13.89% | +0.86% | +13.31%| +4.78%  |
>
> > We thank the reviewer for raising this important point. We agree that standard reasoning benchmarks mainly evaluate final-answer correctness and do not directly reveal whether Stage-1 preserves the content and quality of reasoning traces. To address this, we follow the framework in [5] proposing two cot-level metrics with GPT-4o as the judge:
> >
> > * Reasoning Internal Coherence (**RIC**): how logically consistent and self-contained the reasoning trace is.
> > * Reasoning–Answer Alignment (**RAA**): how well the reasoning trace semantically supports the final answer.
> >
> > We score the LRM and our merged model on three datasets: IFEval, AIME25, and GPQA. For each sample, we provide the question, the ground-truth, the full reasoning trace, and the answer response, and obtain 0–5 scores for RIC and RAA. The results are summarized in Tab.R5 and Tab.A12 (Appendix J.9). We observe that on IFEval, RAIN-Merging slightly improves both RIC and RAA over the LRM. On the reasoning benchmarks AIME25 and GPQA, RAIN-Merging achieves substantial gains in RAA ($>+13\%$) while keeping RIC.
> >
> > These results indicate that stage-1 not merely preserves the surface-level format of reasoning, but also maintains the coherence of the underlying reasoning traces. Crucially, these traces become more faithfully aligned with the final answers. This also helps explain why we observe improvements on reasoning/general benchmarks, as stronger instruction-following makes the model more likely to follow the intended reasoning strategy, which in turn improves reasoning–answer consistency. We have added this analysis and the experiment to Appendix xxx of the revised paper.
>
> **W3.3: Semantic Robustness to Paraphrased Instructions**
>
> > **Table R6**: Robustness to paraphrased instructions on IFEval. We merge the Qwen2.5-7B family under the same configuration as Tab.1 in manuscript. **Acc** is hard accuracy (\%) of satisfying all instruction constraints. **Robustness** is defined as $\text{Acc}({\text{paraphrase})} / \text{Acc}({\text{original})}$.
>
> | Model  | IFEval-original Acc | IFEval-paraphrase Acc | Robustness |
> |-|-|-|-:|
> | Qwen2.5-7B-Instruct          | 65.71             | 64.16                  | 0.98       |
> | DeepSeek-R1-Distill-Qwen-7B  | 57.86             | 50.85                  | 0.88       |
> | Qwen2.5-7B-RAIN-Merging      | 61.29             | 61.81                  | 1.01       |
>
> > Thank you for pointing out this important aspect.
> >
> > First, we note that two instruction-following benchmarks in our experiments already include non-trivial semantic evaluation components. InfoBench and ComplexBench explicitly evaluate whether the content of the response aligns with the prompt, not just simple pattern matching. Thus some degree of semantic evaluation is already baked into their metrics.
> >
> > Moreover, to directly address your concern, we additionally follow IFEval-extended[6] and construct a paraphrased IFEval set. Concretely, we select 200 valid IFEval examples (IFEval-original) and use GPT-4o to generate three paraphrases for each instruction, yielding 600 phrased instructions (IFEval-paraphrase). We then evaluate the hard accuracy on the original IFEval subset and IFEval-paraphrase. We also propose $\textbf{Robustness}=\text{Acc}({\text{paraphrase})} / \text{Acc}({\text{original})}$ to measure how well performance is preserved under paraphrasing.
> >
> > The results are shown in Tab.R6 and Tab.A13 in Appendix J.11. We observe that the LRM’s performance degrades notably under paraphrasing, indicating sensitivity to the exact wording of instructions. Our merged model improves the LRM's accuracy both on the original and paraphrased instructions with a robustness of 1.01, surpassing both the LRM and the ITM. These results suggest that RAIN-Merging strengthens semantic instruction following rather than merely exploiting specific phrasings. We have added this paraphrase-robustness experiment and discussion to the revised Appendix J.11.

---

> ### Author Response · Authors · 2025-11-21
> **Response to Reviewer i2ye (Part 3)**
>
> **W3.2: Instruction-Type Breakdown on IFEval**
>
> > We thank the reviewer for pointing out the need for more fine-grained metrics. To provide a more detailed view, we conduct an instruction-type analysis on IFEval. For each type, we compute the accuracy of the LRM (DeepSeek-R1-Distill-Qwen-7B) and of our merged model (Qwen2.5-7B-RAIN-Merging), and visualize the comparison in Fig.A7 of Appendix J.10. The results show that RAIN-Merging yields the largest gains on constraint types such as `response_language`, `number_words`, and `no_comma`. For most other instruction types, RAIN-Merging either matches or modestly improves the LRM’s performance. We have added the plot and discussion to the revised Appendix J.10 to complement the aggregate accuracy metrics.
>
> **W4.2&Q3: Handling Highly Entangled Instruction and Reasoning Samples in Stage-2**
>
> > We appreciate the reviewer’s insightful questions regarding the entanglement of reasoning and instruction-following in stage-2. We would like to clarify two aspects of how this relates to our method.
> >
> > **Highly entangled scenarios are not ideal for instruction calibration data.** First, the purpose of the instruction calibration set is to ensure that attention can be properly aligned with instruction spans, and to clearly identify which modules correspond to instruction-following. Including heavily entangled examples would make it difficult to distinguish the contribution of instruction-following from reasoning, and lead to unreliable attention-guided scaling. Therefore, we manually screen the calibration data to ensure that instructions and reasoning are clearly separated.
> >
> > **Generalization of stage-2 to entangled tasks.** Second, we agree that in some complex tasks, reasoning is closely linked to instruction-following. As demonstrated in previous **W2**, **MathIF** is a benchmark where mathematical reasoning and instruction constraints are highly entangled in a single task, yet our method still yields substantial gains on the joint “Both Acc” metric (correct answers *and* constraints) in Tab.R4. This suggests that stage-2, though calibrated on disentangled instructions, can generalize effectively to settings where reasoning and instruction-following are tightly coupled.
> >
> > We also believe that exploring calibration procedures that directly model highly intertwined instruction–reasoning signals is an interesting direction for future work.
>
> **Q1: Hypothesis on Why Reasoning and Instruction-Following Are Weakly Coupled**
> > We thank the reviewer for this thoughtful question. Our empirical analysis shows that the task vectors of the LRM and the ITM occupy largely orthogonal principal subspaces in several key modules, suggesting a low degree of coupling in parameter space. We view this as evidence that merging the two capabilities is feasible, while we agree that understanding why this happens is a deeper and more fundamental question.
> >
> > Our current hypothesis is that the low coupling arises from a combination of structural and semantic factors:
> >
> > * For structural separation, LRMs such as R1 are trained with explicit thinking segments (`<think>…</think>`) and separated answers, whereas ITMs primarily optimize the final response.
> > * For semantic specialization, the underlying objectives of LRMs and ITMs are also semantically distinct. Reasoning data emphasize step-by-step logical inference, mathematical computation, and consistency of the internal chain, while instruction-tuning data emphasize following explicit constraints, styles, and user-specified formats.
> >
> > Our current evidence of low coupling in parameter space for LRMs and ITMs is primarily empirical but not coincidental, as demonstrated by the consistent patterns observed across multiple scales. We do not claim a complete theoretical account linking structural and semantic factors to the observed parameter geometry. Instead, we view the formalization of this relationship between data domain and task-vector geometry as an open problem and a key future direction.
>
> ### Reference:
>
> [1] Pyatkin, Valentina, et al. "Generalizing Verifiable Instruction Following." NeurIPS 2025 Track on Datasets and Benchmarks.
>
> [2] Li, Zhenyu, et al. "Xifbench: Evaluating large language models on multilingual instruction following." NeurIPS 2025 Track on Datasets and Benchmarks.
>
> [3] Zou, Tao, et al. "EIFBENCH: Extremely Complex Instruction Following Benchmark for Large Language Models." EMNLP 2025.
>
> [4] Fu, Tingchen, et al. "Scaling reasoning, losing control: Evaluating instruction following in large reasoning models." arXiv 2025.
>
> [5] Wang, Changyue, et al. "Joint Evaluation of Answer and Reasoning Consistency for Hallucination Detection in Large Reasoning Models." arXiv 2025.
>
> [6] Kovalevskyi, Bohdan. "Ifeval-extended: Enhancing instruction-following evaluation in large language models through dynamic prompt generation." Journal of Artificial Intelligence General science 2024.

---

> > ### Comment · Reviewer_i2ye · 2025-11-25
> >
> > I thank the authors for their detailed response. As most of my concerns have been addressed, I have raised my score.

---

> > > ### Author Response · Authors · 2025-11-25
> > >
> > > Thank you very much for your thoughtful follow-up and for taking the time to reconsider our submission. We are glad that our responses have addressed your concerns, and we sincerely appreciate your raised score and positive evaluation.

---

### Official Review · Reviewer_snDt · 2025-11-01

**Soundness:** 4
**Presentation:** 4
**Contribution:** 4
**Rating:** 8
**Confidence:** 5

**Summary:**

This paper presents RAIN-Merging, a novel two-stage, gradient-free method for merging Large Reasoning Models (LRMs) with Instruction-Tuned Models (ITMs) to enhance instruction-following capability while preserving structured reasoning outputs. The approach leverages task-vector orthogonality and introduces a reasoning-aware null-space projection to maintain thinking formats, combined with instruction-attention guided coefficients to improve instruction adherence. Extensive experiments across multiple benchmarks and model scales demonstrate that RAIN-Merging significantly improves instruction-following performance without compromising reasoning or general capabilities. The method offers a computationally efficient and interpretable alternative to supervised fine-tuning, making it highly relevant for real-world applications of LRMs.

**Strengths:**

1.   Novel Research Problem: The work addresses an important and underexplored challenge—balancing instruction-following and reasoning capabilities in LRMs—through model merging, a lightweight and training-free approach.
2.  Effective Methodology: The two-stage RAIN-Merging framework is well-motivated, combining null-space projection to preserve reasoning structure with attention-guided scaling to enhance instruction alignment, all without gradient updates.
3.   Comprehensive Experiments: The paper provides extensive evaluations across multiple instruction-following, reasoning, and agentic benchmarks, with consistent improvements shown across different model sizes and architectures.
4.  Clear and Well-Structured Writing: The paper is clearly written, with a logical flow, detailed derivations, and accessible explanations of both the motivation and technical contributions.

**Weaknesses:**

While the method is evaluated on several model families (Qwen, Llama), further validation on a wider range of architectures and modalities (e.g., multimodal or multilingual models) would strengthen the generalizability claims.

**Questions:**

1.  Overall, I think you have done very meaningful work. My question is: Will your work be open-sourced as a toolkit in the future? I am very much looking forward to using the methods proposed in your paper.
2.  Have you considered validating your method on larger model sizes (e.g., above 30B parameters) to further verify its effectiveness?

---

> ### Author Response · Authors · 2025-11-21
> **Response to Reviewer snDt**
>
> Thank you for your constructive feedback and positive assessment of our work's novelty, methodological effectiveness, comprehensive evaluations, and clear presentation. Below we address your concerns in turn:
>
> **W1: Generalization to Other Architectures and Modalities**
>
> > Thank you for this very helpful suggestion. We agree that validating RAIN-Merging on a wider range of architectures and modalities would further strengthen our generalization claims. A thorough exploration along these lines would require carefully disentangling instruction-following and reasoning and building calibration sets for each setting, which we view as substantial and exciting future work and have already pointed out in Appendix L (iv).
>
> **Q1: Open-Sourcing as Toolkit**
>
> > Thank you very much for the encouraging feedback. We fully share this goal: we are organizing our implementation and plan to open-source as a toolkit soon after the review period, so that other researchers and practitioners can directly reproduce our results and apply RAIN-Merging to new LRM–ITM pairs.
>
> **Q2: Validation on Larger Models**
>
> > **Table R2**: Merging performance and relative gains of RAIN-Merging on the Qwen2.5-32B family under the same configuration as Tab.2 in manuscript. The subsequent “(*relative gain*)” row reports the relative improvement of our method over the LRM.
>
> | Model | IFEval | CELLO | InfoBench | ComplexBench | Average | Math | GPQA | Aider | Arena-Hard-v2 | Average |
> |-------|--------|-------|--------|----------|-----|------|------|-------|---------|-----|
> | Qwen2.5-32B-Instruct | 78.56 | 18.59 | 84.40 | 46.91 | 57.11 | 52.35 | 36.87 | 57.78 | 81.90 | 57.22 |
> | DeepSeek-R1-Distill-Qwen-32B | 76.52 | 19.69 | 83.56 | 44.44 | 56.05 | 68.00 | 60.10 | 54.81 | 82.00 | 66.23 |
> | Qwen2.5-32B-RAIN-Merging | 77.26 | 19.96 | 84.76 | 45.74 | 56.93 | 75.67 | 61.62 | 54.07 | 83.70 | 68.77 |
> | (*relative gain*) | +0.97% | +1.39% | +1.44% | +2.93% | +1.57% | +11.28% | +2.52% | -1.35% | +2.07% | +3.83% |
>
> > Thank you for this suggestion. We agree that assessing RAIN-Merging at larger scales is important for understanding its practical impact. Following your comment, we have conducted additional experiments on the Qwen2.5-32B family, using DeepSeek-R1-Distill-Qwen-32B as the LRM and Qwen2.5-32B-Instruct as the ITM.
> >
> > The new results are reported in Tab.R2 and Tab.A9 in Appendix J.6 of the revised paper. Compared to the 32B LRM, our Qwen2.5-32B-RAIN-Merging improves instruction-following performance on all four instruction-following benchmarks with comparable performance on reasoning and general ability. These results confirm that our RAIN-Merging remains effective at the 32B level.

---

### Official Review · Reviewer_Yy8R · 2025-11-02

**Soundness:** 3
**Presentation:** 2
**Contribution:** 3
**Rating:** 6
**Confidence:** 3

**Summary:**

The paper proposes RAIN-Merging, a training-free (gradient-free) technique to improve instruction following in Large Reasoning Models (LRMs) without degrading their explicit “thinking→answer” format. The method has two stages: (1) Reasoning-aware null-space projection that projects the instruction-tuned model’s task vector into the null space of forward features at special thinking tokens (e.g., <think> … </think>), thereby preserving the LRM’s thinking distribution (formalized via a KL constraint). (2) Instruction-attention guided scaling, which uses forward attention statistics from a small instruction calibration set to compute per-module coefficients that increase attention alignment to instruction spans while penalizing leakage to unrelated spans. Experiments across Qwen/Llama backbones and sizes show instruction-following gains with maintained or improved reasoning and lower compute than SFT; ablations support both stages’ roles and show the method keeps the <think> format intact (no missing terminator).

**Strengths:**

Clear problem & neat insight. The paper pinpoints a real pain point: LRMs reason well, but violate format/constraints. The idea to protect the thinking segment explicitly while injecting instruction-following behavior is crisp and well-motivated.

Strong empirical results. On the headline 7B setting, RAIN-Merging improves instruction-following average (48.11 vs. 44.12 LRM; +4 points absolute) while also improving reasoning/general (55.59 vs. 51.03) and beating task-arithmetic, SLERP, Karcher, TIES, DARE-TIES, and activation-based methods (AIM/ACM/LEWIS combined with TIES)

Efficiency. Minutes to merge (20.96 min reported) vs. SFT’s 120+ min; GPU memory also far below SFT (22.1 GB vs. 112.6 GB in their config)

**Weaknesses:**

Calibration-set specificity. The instruction calibration set is distilled from IFEval-style instructions (365 samples). This may bias the proxy to rule-verifiable patterns and possibly underrepresent open-ended or tool-use instructions.

Reliance on explicit thinking markers. Stage 1 presumes accessible special tokens and feature extraction around them. It is unclear how well this transfers to LRMs with different templates (or hidden/implicit thinking) or to models without consistent <think> tags, using ReAct format for thinking tool use.

**Questions:**

How does Stage 1 handle LRMs whose “thinking” is not demarcated by explicit tokens, or that interleave tool calls and thoughts (e.g., ReAct-style)?

Generalization of the proxy. If the calibration set used open-ended instructions (no machine-checkable rules), would Stage 2 still pick effective heads? Any results using other instruction corpora?

---

> ### Author Response · Authors · 2025-11-21
> **Response to Reviewer Yy8R**
>
> Thank you for your constructive feedback and for recognizing our work's core insight and strong empirical results. Below we address your concerns in turn:
>
> **W1&Q2: Instruction Calibration Set Generalization.**
>
> > **Table R1**: Performance of RAIN-Merging with different instruction calibration sets in Stage-2. **Rule** is our original instruction calibration set from IFEval, **Open** is the new set from InforBench, **Rule+Open** simply concatenates both sets. We merge Qwen2.5-7B-Instruct (ITM) into DeepSeek-R1-Distill-Qwen-7B (LRM) under the same configuration as Tab.1 in manuscript.
>
> | Method | IFEval | CELLO | InfoBench | ComplexBench | Avg. | Math | GPQA | Aider | Arena-Hard-v2 | Avg. |
> |--------|--------|-------|--------|----------|-----|------|------|-------|---------|-----|
> | LRM | 55.45 | 16.59 | 71.73 | 32.73 | 44.12 | 64.75 | 44.44 | 29.63 | 65.29 | 51.03 |
> | RAIN-Merging (Rule) | 63.22 | 19.03 | 74.53 | 35.66 | 48.11 | **68.75** | **54.55** | 33.33 | **65.73** | **55.59** |
> | RAIN-Merging (Open) | 62.92 | 19.24 | 74.89 | 35.67 | 48.15 | 65.14 | 49.49 | 31.11 | 64.67 | 52.59 |
> | RAIN-Merging (Rule+Open) | **64.03** | **19.63** | **75.64** | **36.70** | **49.00** | 67.43 | 53.03 | **35.56** | 65.29 | 55.32 |
>
> > We thank the reviewer for raising the concern that our instruction calibration set is only distilled from rule-verifiable instructions of IFEval, which might bias our stage-2 proxy and underrepresent open-ended instructions. To clarify this, we constructed a new open-ended calibration set from InfoBench (e.g., constraints on tone, style, content focus) following the same pipeline as in Appendix H.2 with manual filtering. This yields a 260-example open-ended instruction calibration set (Open), complementary to the original 365-example rule-verifiable set from IFEval (Rule).
> >
> > We then evaluate on the Qwen2.5-7B LRM–ITM pair with three calibration variants: Rule (original), Open, and Rule+Open (simply concatenating both sets). The results are summarized in Tab.R1 and Tab.A8 (Appendix J.5).
> > * Using Open alone slightly improves on open-ended benchmarks (InfoBench, ComplexBench). This shows that the instruction-attention guided coefficients continue to pick effective heads even when calibrated exclusively on open-ended instructions.
> > * The mixed Rule+Open calibration further improves all four instruction-following benchmarks simultaneously, raising the average from 48.11 to 49.00, indicating that combining rule-verifiable and open-ended instructions yields a more generalized proxy.
> > * However, a purely open-ended calibration and the Rule+Open variant incur a drop in reasoning/general performance, suggesting that we need to carefully weigh the improvement in instruction-following by open-ended instructions against the potential impairment of reasoning abilities.
> >
> > We now include the above results and discussion in the revised Appendix J.5 and believe they demonstrate that stage-2 is not only tied to IFEval-style rule-verifiable instructions, but can also be driven by open-ended corpora. Designing even cleaner open-ended calibration sets and better handling instruction–reasoning entanglement is an interesting direction for future work.
>
> **W2&Q1: Applicability of Stage-1 Beyond Explicit Thinking Tokens**
>
> > We appreciate the reviewer’s concern. We would like to clarify that **Stage-1 is mathematically applicable to any input token**. Concretely, the null-space projection $P^{\perp}(\cdot)$ in Eq.(5) can be applied to forward features $\Phi^k_{\Omega}$ on any subset of token indices $\Omega$, not only explicit thinking markers.
> >
> > For these LRMs without explicit thinking tokens or with ReAct-style traces, a straightforward choice is to take the entire reasoning span and apply the same null-space projection on those features. More refined variants require further analysis to account for the impact of merging on these thinking traces output. Once such a subset is specified, stage-1 can be directly applied to effectively control the output distribution shift of these tokens. We believe that extending to implicit reasoning or ReAct-style LRMs is an interesting direction for future work.

---

### Author Response · Authors · 2025-11-21
**General Response**

Dear Program Chairs, Senior Area Chairs, Area Chairs, and Reviewers,

We sincerely appreciate your time, constructive critiques, and insightful suggestions, which have substantially strengthened our work. We are particularly grateful for the reviewers' recognition of our method's effectiveness and their positive assessment of our motivation.

In response to the comments, we have addressed each point carefully and conducted extensive additional experiments to further validate the scalability, generalization, robustness, and internal mechanisms of RAIN-Merging. All modifications in the revised PDF have been highlighted in blue for ease of reference.

**Additional Experiments: (Weakness or Question, Table in Rebuttal, *Table or Figure, Revision in Revised PDF*)**

1. Generalization and Scalability
    * Generalization of Instruction Calibration Sets (Reviewer Yy8R W1&Q2, Tab.R1, *Tab.A8, Appendix J.5*)
    * Validation on Larger Models (Reviewer snDt Q2, Tab.R2, *Tab.A9, Appendix J.6*)
    * Generalization to Unseen Benchmarks (Reviewer i2ye W1, Tab.R3, *Tab.A10, Appendix J.7*)
2. In-depth Analysis of Instruction-Following and Reasoning
    * Joint Capability Evaluation (Reviewer i2ye W2, Tab.R4, *Tab.A11, Appendix J.8*)
    * CoT Quality Analysis (Reviewer i2ye W3.1&W4.1&Q2, Reviewer 2pjo W1, Tab.R5 & R7, *Tab.A12, Appendix J.9*)
    * Fine-grained Instruction Analysis (Reviewer i2ye W3.2, *Fig.A7, Appendix J.10*)
    * Case Studies (Reviewer 2pjo W1, *Page 37-40, Appendix J.13*)
3. Robustness and Ablation
    * Semantic Robustness to Instructions (Reviewer i2ye W3.3, Tab.R6, *Tab.A13, Appendix J.11*)
    * Complete Ablation Study (Reviewer 2pjo W2, Tab.R8, *Tab.4, Sec.4.2 RQ3*)

**Clarification: (Weakness or Question)**

1. Beyond Explicit Thinking Tokens (Reviewer Yy8R W2&Q1)
2. More Modalities and Open-Sourcing (Reviewer snDt W1&Q1)
3. Highly Entangled Instruction and Reasoning (Reviewer i2ye W4.2&Q3)
4. Hypothesis on Weakly Coupled Parameter Space (Reviewer i2ye Q1)

We welcome any further questions or suggestions from the reviewers and look forward to continued discussion.

---

### Author Response · Authors · 2025-12-02
**Summary of Discussion and Rebuttal Updates**

Dear Program Chairs, Senior Area Chairs, Area Chairs, and Reviewers,

We sincerely thank all reviewers for their constructive feedback and appreciate the time and effort dedicated to evaluating our work. In our rebuttal, we have actively and systematically responded to every point raised in the reviews. We have provided detailed clarifications, conducted additional analyses and experiments, and made corresponding revisions to the manuscript in an effort to fully respond to the reviewers' concerns.

Below, we provide a point-by-point summary of our responses to each reviewer’s comments.

* **Reviewer i2ye (Initial Rating 4, raise score):** We addressed concerns regarding further in-depth validation effectiveness through comprehensive additional experiments and clarifications, such as more benchmarks, CoT quality evaluation, robustness to paraphrasing, and instruction-reasoning entanglement. The reviewer acknowledged our rebuttal, stating **"most of my concerns have been addressed,"** and explicitly **raised score**.

* **Reviewer 2pjo (Initial Rating 8, maintain score):** We incorporated the requested ablation studies and a detailed case study on **GPQA** into the revised paper. The reviewer praised the new results as **"very interesting"** and effective, concluding with **"Great work!"** and maintaining positive assessment.

* **Regarding Reviewer Yy8R (Initial Rating 6, no follow-up):** We conducted additional experiments to verify generalization of instruction calibration set and clarified the applicability of null-space projection.

* **Regarding Reviewer snDt (Initial Rating 8, no follow-up):** We provided the suggested validation on larger models. The results confirm that our method's performance scales with model size.

**Note on Revision:** All changes, including the experiments and clarifications, have been highlighted in blue in the revised PDF for easy tracking.

We sincerely hope this consolidated summary facilitates your evaluation by clearly demonstrating how all reviewer concerns have been thoroughly addressed. We are deeply grateful for the time and thoughtful consideration given by both the reviewers and the ACs. In particular, we would like to express our special appreciation for your dedicated efforts in managing the review process under this year's extraordinary review circumstances.

Sincerely,

Authors

---

### Meta-Review · Area_Chair_mgb1 · 2025-12-21

**Summary:**

See below.

**Reviewer Concerns:**

The authors provided a clear synthesis of what happended during the discussion.
I copy-paste it:

Below, we provide a point-by-point summary of our responses to each reviewer’s comments.

* Reviewer i2ye (Initial Rating 4, raise score): We addressed concerns regarding further in-depth validation effectiveness through comprehensive additional experiments and clarifications, such as more benchmarks, CoT quality evaluation, robustness to paraphrasing, and instruction-reasoning entanglement. The reviewer acknowledged our rebuttal, stating "most of my concerns have been addressed," and explicitly raised score.

* Reviewer 2pjo (Initial Rating 8, maintain score): We incorporated the requested ablation studies and a detailed case study on GPQA into the revised paper. The reviewer praised the new results as "very interesting" and effective, concluding with "Great work!" and maintaining positive assessment.

* Regarding Reviewer Yy8R (Initial Rating 6, no follow-up): We conducted additional experiments to verify generalization of instruction calibration set and clarified the applicability of null-space projection.

* Regarding Reviewer snDt (Initial Rating 8, no follow-up): We provided the suggested validation on larger models. The results confirm that our method's performance scales with model size.

**Reviewer Scores:**

see above.

---

### Decision · Program_Chairs · 2026-01-26

Accept (Oral)